# Atomic fluctuations lifting the energy degeneracy in Si/SiGe quantum dots

Brian Paquelet Wuetz[1,6], Merritt P. Losert[2,6], Sebastian Koelling[3,6], Lucas E. A. Stehouwer[1], Anne-Marije J. Zwerver[1], Stephan G. J. Philips[1], Mateusz T. Mądzik[1], Xiao Xue[1], Guoji Zheng [1], Mario Lodari[1], Sergey V. Amitonov[1], Nodar Samkharadze[4], Amir Sammak[4], Lieven M. K. Vandersypen [1], Rajib Rahman [5], Susan N. Coppersmith [5], Oussama Moutanabbir [3], Mark Friesen [2] & Giordano Scappucci [1] ✉

Electron spins in Si/SiGe quantum wells suffer from nearly degenerate conduction band valleys, which compete with the spin degree of freedom in the formation of qubits. Despite attempts to enhance the valley energy splitting deterministically, by engineering a sharp interface, valley splitting fluctuations remain a serious problem for qubit uniformity, needed to scale up to large quantum processors. Here, we elucidate and statistically predict the valley splitting by the holistic integration of 3D atomic-level properties, theory and transport. We find that the concentration fluctuations of Si and Ge atoms within the 3D landscape of Si/SiGe interfaces can explain the observed large spread of valley splitting from measurements on many quantum dot devices. Against the prevailing belief, we propose to boost these random alloy composition fluctuations by incorporating Ge atoms in the Si quantum well to statistically enhance valley splitting.

Advanced semiconductor manufacturing is capable of integrating billions of transistors onto a single silicon chip. The promise of leveraging the same technology for large-scale integration of qubits into a fault-tolerant quantum processing unit is a key driver for developing electron spin qubits in silicon quantum dots[1]. Although these devices bear many similarities to transistors[2], qubits operate in the single electron regime[3], making them more sensitive to electrostatic disorder and noise arising from the surrounding environment. In strained silicon quantum wells, the electronically active part of the device is separated by an epitaxial SiGe barrier from the electronically noisy interface at the gate-stack, offering a quiet system with high mobility and low leakage between the gate and the quantum dots[4]. These properties make strained Si/SiGe heterostructures promising for scalable qubit tiles[5,6] and have made it possible to define nine quantum dot arrays[7], run quantum algorithms[8] and entangle three-spin states[9] in

natural silicon structures, and achieve two-qubit gate fidelity above 99%[10,11] in isotopically purified silicon structures.

However, spin-qubits in silicon suffer from a two-fold degeneracy of the conduction band minima (valleys) that creates several non-computational states that act as leakage channels for quantum information[12]. These leakage channels increase exponentially with the qubit count[13], complicating qubit operation and inducing errors during spin transfers. Despite attempts to enhance the valley energy splitting, the resulting valley splittings are modest in Si/SiGe heterostructures, with typical values in the range of 20 to 100 μeV[8,14–20] and only in a few instances in the range of 100 to 300 μeV[21–23]. Such variability in realistic silicon quantum dots remains an open challenge for scaling to large qubit systems. In particular, the probability of thermally occupying the excited valley state presents a challenge for spin initialization, and, in some cases, intervalley scattering may limit

[1]QuTech and Kavli Institute of Nanoscience, Delft University of Technology, Delft, The Netherlands. [2]University of Wisconsin-Madison, Madison, WI, USA. [3]Department of Engineering Physics, École Polytechnique de Montréal, Montréal, Québec, Canada. [4]QuTech and Netherlands Organisation for Applied Scientific Research (TNO), Delft, The Netherlands. [5]University of New South Wales, Sydney, Australia. [6]These authors contributed equally: Brian Paquelet Wuetz, Merritt P. Losert, Sebastian Koelling. ✉e-mail: g.scappucci@tudelft.nl

the spin coherence[24]. Furthermore, small valley splitting may affect Pauli spin blockade readout[25], which is considered in large-scale quantum computing proposals[5,6]. Therefore, scaling up to larger systems of single-electron spin qubits requires that the valley splitting of all qubits in the system should be much larger than the typical operation temperatures (20−100 mK).

It has been known for some time that valley splitting depends sensitively on the interface between the quantum well and the SiGe barrier[26]. Past theoretical studies have considered disorder arising from the quantum well miscut angle[27] and steps in the interface[28–32] demonstrating that disorder of this kind can greatly decrease valley splitting in quantum dots. However, a definitive connection to experiments has proven challenging for a number of reasons. At the device level, a systematic characterisation of valley splitting in Si/SiGe quantum dots has been limited because of poor device yield associated with heterostructure quality and/or device processing. At the materials level, atomic-scale disorder in buried interfaces[33] may be revealed by atom-probe tomography (APT) in three-dimensions (3D) over the nanoscale dimensions comparable to electrically defined quantum dots. However, the current models employed to reconstruct in 3D the APT data can be fraught with large uncertainties due to the assumptions made to generate the three-dimensional representation of the tomographic data[34]. This results in limited accuracy when mapping heterointerfaces[35] and quantum wells[36–38]. These limitations prevent linking the valley splitting in quantum dots to the relevant atomic-scale material properties and hinder the development of accurate and predictive theoretical models.

Herein we solve this outstanding challenge and establish comprehensive insights into the atomic-level origin of valley splitting in realistic silicon quantum dots. Firstly, we measure valley splitting systematically across many quantum dots, enabled by high-quality heterostructures with a low disorder potential landscape and by improved fabrication processes. Secondly, we establish a new method to analyse APT data leading to accurate 3D evaluation of the atomic-level properties of the Si/SiGe buried interfaces. Thirdly, incorporating the 3D atomic-level details obtained from APT, we simulate valley splitting distributions that consider the role of random fluctuations in the concentration of Si and Ge atoms at each layer of the Si/SiGe interfaces. By comparing theory with experiments, we find that the measured random distribution of Si and Ge atoms at the Si/SiGe interface is enough to account for the measured valley splitting spread in real quantum dots. Based on these atomistic insights, we conclude by proposing a practical strategy to statistically enhance valley splitting above a specified threshold as a route to making spin-qubit quantum processors more reliable−and consequently−more scalable.

## Results
### Material stacks and devices
Figure 1 overviews the material stack, quantum dot devices, and measurements of valley splitting. To increase statistics, we consider two isotopically purified $^{28}$Si/Si$_{0.7}$Ge$_{0.3}$ heterostructures (quantum wells A and B) designed with the same quantum well width and top-interface sharpness (Methods), which are important parameters determining valley splitting[23,26]. As shown in high angle annular dark field scanning transmission electron microscopy (HAADF-STEM),

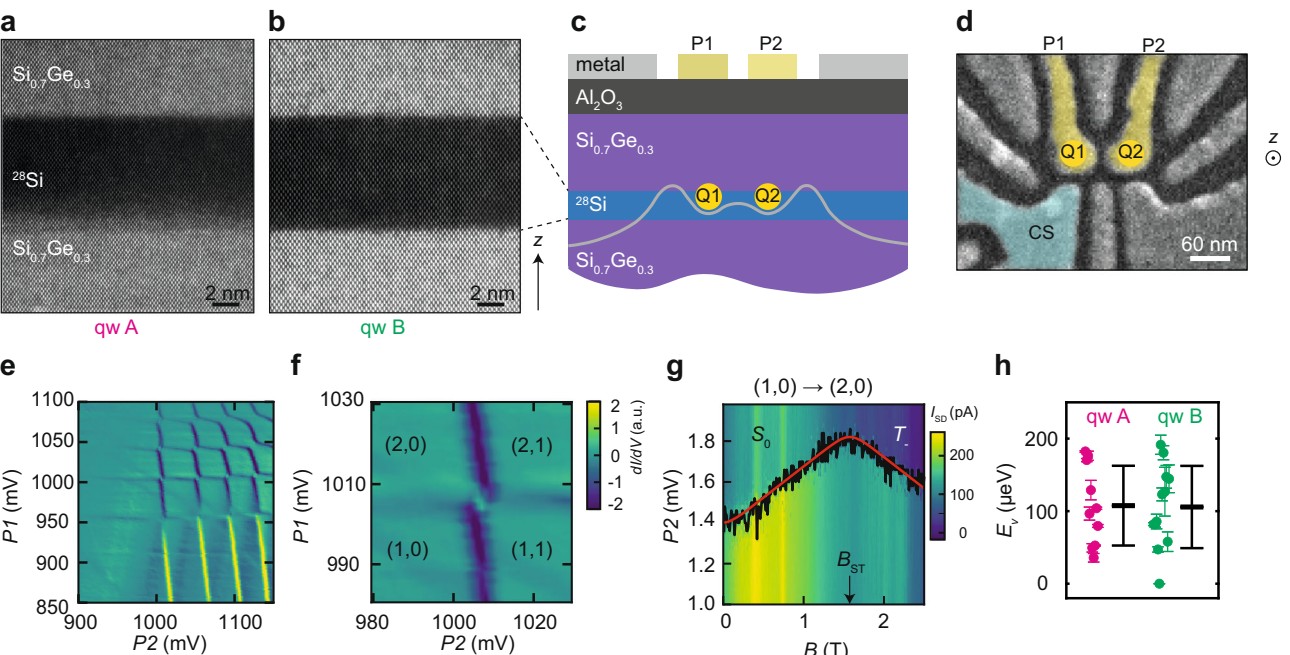

**Fig. 1 | Material stack, devices, and valley splitting measurements. a, b** High-angle annular dark field scanning transmission electron microscopy (HAADF-STEM) of $^{28}$Si/SiGe quantum wells A and B, respectively. **c, d** Schematic cross-section of a heterostructure with gate layout and false-coloured scanning electron microscope image of a double quantum dot, respectively. Q1 and Q2 are the quantum dots defined through confinement potentials (schematic, grey line) formed below plunger gates P1 and P2. CS is a nearby quantum dot used as a charge sensor. **e** Typical stability diagram of a double quantum dot formed by plunger gates P1 and P2 and measured by a nearby charge sensor (CS in **d**). **f** Close-up of the stability diagram in the few-electron regime. **g** Typical magnetospectroscopy of the (1,0) → (2,0) transition, used to measure singlet-triplet splittings. An offset of 1082 mV is subtracted for clarity from the gate voltage applied to P2. Black lines show the

location of the maximum of the differentiated charge-sensor signal ($dI_{SD}/dP2$) of the electron charging transition. Red lines show a fit to the data, from which we extract the kink position $B_{ST}$. The valley splitting $E_v$ is given by $g\mu_B B_{ST}$, where $g = 2$ is the gyromagnetic ratio and $\mu_B$ is the Bohr magneton. **h** Experimental scatter plots of the valley splittings for quantum wells A (magenta) and B (green), with thick and thin horizontal black lines denoting the mean and two-sigma error bars. For quantum well B, the data point $E_V = 0$ μeV indicates that the kink in magnetospectroscopy associated with valley splitting was not observed and, consequently, that the valley splitting is below the lower bound of about 23 μeV set by our experimental measurement conditions (see Supplementary Fig. 6 and Supplementary Table 1).

quantum well A (Fig. 1a) has a sharp $^{28}$Si → Si-Ge heterointerface at the top and a diffused Si-Ge → $^{28}$Si heterointerface at the bottom, whereas in quantum well B (Fig. 1b) the growth process was optimized to achieve sharp interfaces at both ends of the quantum well. These heterostructures support a two-dimensional electron gas with high mobility and low percolation density (Supplementary Figs. 1 and 2), indicating a low disorder potential landscape, and high-performance qubits[10,39] with single- and two-qubit gates fidelity above 99%[10].

We define double-quantum dots electrostatically using gate layers insulated by dielectrics (Methods). A positive gate voltage applied to plunger gates P1 and P2 (Fig. 1c) accumulates electrons in the buried quantum well, while a negative bias applied to other gates tunes the confinement and the tunnel coupling between the quantum dots Q1 and Q2. All quantum dots in this work have plunger gate diameters in the range of 40–50 nm (Fig. 1d and Supplementary Table 1), setting the relevant lateral length scale for atomic-scale disorder probed by the electron wave function.

## Valley splitting measurements

We perform magnetospectroscopy measurements of valley splitting $E_v$ in dilution refrigerators with electron temperatures of about 100 mK (Methods). Figure 1e shows a typical charge stability diagram of a double quantum dot with DC gate voltages tuned to achieve the few electron regime, highlighted in Fig. 1f. We determine the 2-electron singlet-triplet energy splitting ($E_{ST}$) by measuring the gate-voltage dependence as a function of parallel magnetic field $B$ along the $(0,1) \rightarrow (0,2)$ transition (Fig. 1g) and along the $(1,1) \rightarrow (0,2)$ transition (Supplementary Fig. 4). In Fig. 1g, the transition line (black line) slopes upward, because a spin ↑ electron is added to form a singlet ground state $S_0$. Alternatively, a spin down electron can be added to form a $T_-$-state, with a downward slope. A kink occurs when the $S_0$-state is energetically degenerate with the $T_-$-state, becoming the new ground state of the two-electron-system. From the position of the kink

$(B_{ST} = 1.57$ T) along the theoretical fit (red line) and the relation $E_{ST} = g\mu_B B_{ST}$, where $g = 2$ is the electron gyromagnetic ratio and $\mu_B$ is the Bohr magneton, we determine $E_{ST} = 182.3$ μeV for this quantum dot. $E_{ST}$ sets a lower bound on the valley splitting, $E_v \geq E_{ST}$[21,40]. Due to small size, our dots are strongly confined with lowest orbital energy much larger than $E_{ST}$ (Supplementary Fig. 3), similar to other Si/SiGe quantum dots[14,18,22]. Therefore, we expect exchange corrections to have negligible effects[40] and here take $E_v \approx E_{ST}$.

Here we report measurements of $E_v$ in 10 quantum dots in quantum well A and 12 quantum dots in quantum well B (Supplementary Figs. 5 and 6) and compare the measured values in Fig. 1h. We observe a rather large spread in valley splittings, however we obtain remarkably similar mean values and two-standard-deviation error bars $\overline{E_v} \pm 2\sigma$ of $108 \pm 55$ μeV and $106 \pm 58$ μeV for quantum wells A and B, respectively. The quantum dots all have a similar design and hence are expected to have similar electric fields across the devices with a small influence on valley splitting under our experimental conditions. We argue that quantum wells A and B have similar $\overline{E_v} \pm 2\sigma$ because the electronic ground state is confined against the top interface, which is very similar in the two quantum wells.

## Atom probe tomography

We now characterise the atomic-scale concentration fluctuations at the quantum well interfaces to explain the wide range of measured valley splittings with informed theoretical and statistical models. To probe the concentrations over the dimensions relevant for quantum dots across the wafer, we perform APT on five samples each from quantum wells A and B, with a field of view of approximately 50 nm at the location of the quantum well (Methods). First, we show how to reliably reconstruct the buried quantum well interfaces, then we use this methodology to characterise their broadening and roughness.

Figure 2a shows a typical point-cloud reconstruction of an APT specimen from quantum well B. Each point represents the estimated

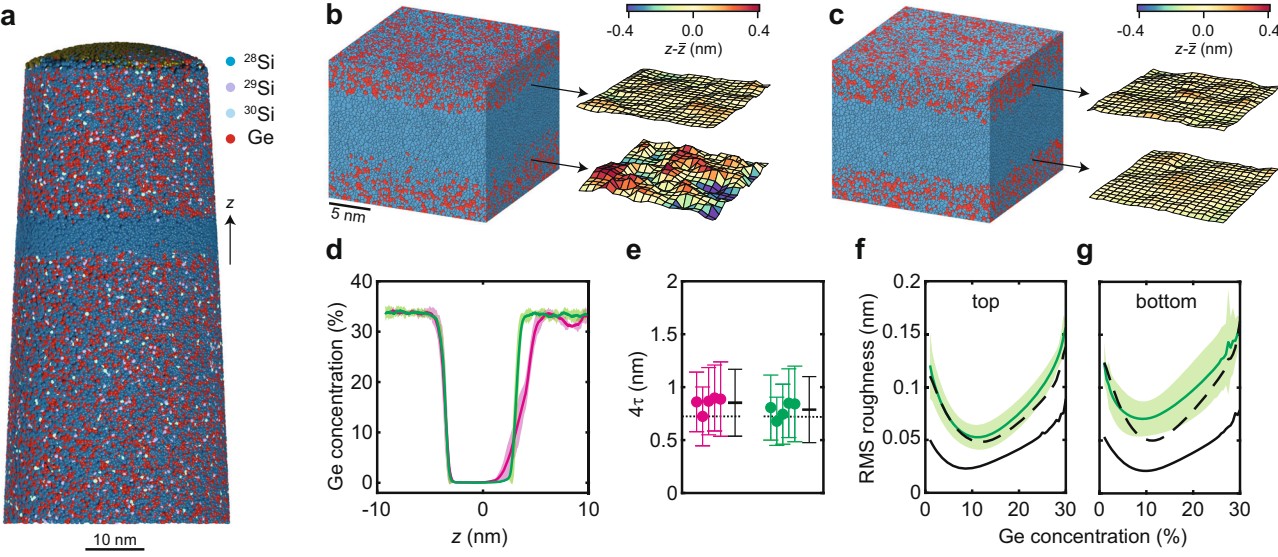

**Fig. 2 | Atom probe tomography of $^{28}$Si/SiGe heterostructures. a** Point-cloud APT reconstruction of quantum well B, showing the $^{28}$Si quantum well and surrounding SiGe barriers. Isotopic purification is confirmed by secondary ion mass spectroscopy (Supplementary Fig. 14). **b, c** Voronoi tessellation of the APT reconstructions for quantum wells A and B, respectively, and extracted isosurfaces corresponding to 8% Ge concentration. $\bar{z}$ is the average position of the 8% Ge concentration across these particular samples. We limit the lateral size of the analysis to ≈30nm × 30 nm, reflecting the typical lateral size of a quantum dot (Fig. 1d). **d** Average germanium concentration depth profiles across quantum wells A (magenta) and B (green). Shaded areas mark the 95% confidence interval over each of the sets of five APT samples. **e** Statistical analysis of the top interface width

$4\tau$ determined by fitting the data for quantum wells A (magenta) and B (green) to sigmoid functions. Thick and thin horizontal black lines denote the mean and two-standard-deviation error bars for the different APT samples. Dotted black lines show $4\tau$ results from the HAADF-STEM measurements (Supplementary Fig 13). **f, g** Root mean square (RMS) roughness of the concentration isosurfaces as a function of germanium concentration at the top and bottom interfaces of quantum well B (green line). Shaded areas indicate the 95% confidence interval, averaged over each set of five APT samples. The experimental data are compared to the RMS roughness of a simulated quantum well with the interface properties of **d** (dashed black line) vs. an atomically sharp quantum well (solid black line).

position of an ionized atom detected during the experiment[34]. Qualitatively, we observe an isotopically enriched $^{28}$Si quantum well, essentially free of $^{29}$Si, cladded in a SiGe alloy. To probe the interface properties with the highest possible resolution allowed by APT and differently from previous APT studies on Si/SiGe[38], we represent the atom positions in the acquired data sets in form of a Voronoi tessellation[41,42] and generate profiles on an $x - y$ grid of the tessellated data, as described in Supplementary Note 2c. A sigmoid function $[1 + \exp(z - z_0)/\tau]^{-1}$[38] is used to fit the profiles of each tile in the $x - y$ grid. Here, $z_0$ is the inflection point of the interface and $4\tau$ is the interface width. As the Voronoi tessellation of the data set does not sacrifice any spatial information, the tiling in the $x - y$ plane represents the smallest lateral length scale over which we characterise the measured disorder at the interface. Note that we do not average at all over the $z$ axis and hence maintain the inherent depth resolution of APT. We find that for tiles as small as 3 nm × 3 nm the numerical fitting of sigmoid functions to the profiles converges reliably. Although each tile contains many atoms, their size is still much smaller than the quantum dot diameter, and may therefore be considered to be microscopic. We use the sigmoid fits for each tile stack to visualise and further characterise the interfaces (Supplementary Figs. 8–10). Importantly, Ge concentration isosurfaces as shown in Fig. 2b, c are constructed by determining the vertical position for which each of the sigmoids reaches a specific concentration. Note, that we oversample the interface to improve the lateral resolution by making the 3 nm × 3 nm tiles partially overlap (Supplementary Note 2c).

In Fig. 2d, we show the average Ge concentration profile and measurement to measurement variations from the tessellated volumes (Supplementary Note 2b, c) of all samples for both quantum wells A and B. APT confirms HAADF-STEM results in Fig. 1a, b: quantum wells A and B have an identical sharp top interface and quantum well A has a broader bottom interface. Furthermore, the shaded colored areas in Fig. 2d reveal narrow 95% confidence levels, pointing to highly uniform concentration profiles when averaged across the wafer. Strong disorder fluctuations emerge at the much smaller tile length scale. In Fig. 2e we show for all samples of a given quantum well the interface width mean value with two standard deviations $\overline{4\tau} \pm 2\sigma$, obtained by averaging over all the tiles in a given sample. The results indicate uniformity of $\overline{4\tau}$, and further averaging across all samples of a given heterostructure ($\mu_{\overline{4\tau}}$, black crosses) yields similar values of $\mu_{\overline{4\tau}} = 0.85 \pm 0.32$ nm and $0.79 \pm 0.31$ nm for quantum wells A and B, consistent with our $4\tau$ analysis from HAADF-STEM measurements (black dotted lines). However, the two-standard-deviation errors ($2\sigma$) of each data point can be up to 30% of the mean value $\overline{4\tau}$.

To pinpoint the root cause of atomic-scale fluctuations at the interface, in Fig. 2f, g we utilize the 3D nature of the APT data sets, calculate, and compare the root mean square (RMS) roughness of the interfaces (solid green lines) as measured by APT on quantum well B to two 3D models (Fig. 2f, g) mimicking the dimensions of an APT data set. Both models are generated with random distributions of Si and Ge in each atomic plane (Supplementary Note 2d). The first model (solid black lines) corresponds to an atomically abrupt interface where the Ge concentration drops from ~33.5% to 0% in a single atomic layer. It hence represents the minimum roughness achievable at each isoconcentration surface given the in-plane randomness of SiGe and the method to construct the interface. The second model (dashed black lines) is generated with the experimentally determined Ge concentration profile along the depth axis (Supplementary Fig. 11). As shown in Fig. 2f, g, the roughness extracted from the second model fits well with the measured data, suggesting that the RMS roughness measured by APT is fully explained by the interface width and shape along the depth axis. Furthermore, as the deviation of each isosurface tile position from the isosurface's average position also matches that of the measured interfaces from the second model (Supplementary Movie 1) the APT data are consistent with a random in-plane

distribution of Ge perpendicular to the interface in all data sets of quantum well B. For 2 out of 5 samples on quantum well A that we analyzed, we observe features that are compatible with correlated disorder from atomic steps (Supplementary Fig. 13). In the following, the alloy disorder observed in the APT concentration interfaces is incorporated into a theoretical model. As shown below, the calculations of valley splitting distributions associated with the 3D landscape of Si/SiGe interfaces can be further simplified into a 1D model that incorporates the in-plane random distribution of Si and Ge atoms.

## Valley splitting simulations

We begin by considering an ideal laterally infinite heterostructure with no concentration fluctuations, and we denote the average Si concentration at layer $l$ by $\bar{x}_l$. Due to the finite size of a quantum dot and the randomness in atomic deposition, there will be dot-to-dot concentration fluctuations. We therefore model the actual Si concentration at layer $l$ by averaging the random alloy distribution weighted by the lateral charge density in the quantum dot, giving $x_l^d = \bar{x}_l + \delta_{x_l}$, as described in Supplementary Note 3c. Here, the random variation $\delta_{x_l}$ is computed assuming a binomial distribution of Si and Ge atoms. We find that these fluctuations can have a significant impact on the valley splitting.

We explore these effects numerically using 1D tight-binding simulations. We begin with the averaged fitted concentration profiles obtained from the APT analysis in Fig. 2d, which enable us to directly measure the average Ge concentration in a given layer $\bar{x}_l$ (Fig. 3a). The variance of the concentration fluctuations is determined by the size of the quantum dot, which we assume has an orbital excitation energy of $\hbar\omega = 4.18$ meV and corresponding radius $\sqrt{\hbar/m^*\omega}$, as well as the average Si concentration $\bar{x}_l$. Here, $m^*$ is the effective mass of Si. Together, $\bar{x}_l$ and the variance determine the probability distribution of weighted Si and Ge concentrations. Concentration profiles are sampled repeatedly from this distribution, with a typical example shown in Fig. 3b. The valley splitting is then determined from a 1D tight-binding model[43]. The envelope of the effective mass wavefunction $\psi_{env}(z)$ is shown in Fig. 3c (grey curve) for an electron confined in the quantum well of Fig. 3b. The procedure is repeated for 10,000 profile samples, obtaining the histogram of valley splittings shown in Fig. 3e. These results agree very well with calculations obtained using a more sophisticated three-dimensional 20-band sp$^3$d$^5$s* NEMO tight-binding model[44] (Supplementary Note 3b) and confirm that concentration fluctuations can produce a wide range of valley splittings. For comparison, at the top of Fig. 3e, we also plot the same experimental valley splittings shown in Fig. 1h, demonstrating good agreement in both the average value and the statistical spread. These observations support our claim that the valley splitting is strongly affected by composition fluctuations due to random distributions of Si and Ge atoms near the quantum well interfaces, even though the experiments cannot exclude the presence of correlated disorder from atomic steps in quantum dots.

Analytical methods using effective mass theory may also be used to characterise the distribution of valley splittings. First, we model the intervalley coupling matrix element[26] as $\Delta = \int e^{-2ik_0 z} U(z) |\psi_{env}(z)|^2 dz$, where $k_0 = 0.82 \times 2\pi/a_0$ is the position of the valley minimum in the Si Brillouin zone, $a_0 = 0.543$ nm is length of the Si cubic unit cell, $\psi_{env}(z)$ is a 1D envelope function, and $U(z)$ is the quantum well confinement potential. The intervalley coupling $\Delta$ describes how sharp features in the confinement potential couple the two valley states, which would otherwise be degenerate. In general, $\Delta$ is a complex number that can be viewed as the sum of two distinct components: a deterministic piece $\Delta_0$, arising from the average interface concentration profile, and a random piece $\delta\Delta$, arising from concentration fluctuations. The latter can be expressed as a sum of contributions from individual atomic layers: $\delta\Delta = \sum_l \delta\Delta_l$, where $\delta\Delta_l$ is proportional to $\delta_{x_l} |\psi_{env}(z_l)|^2$ (see Methods). To visualize the effects of concentration fluctuations in

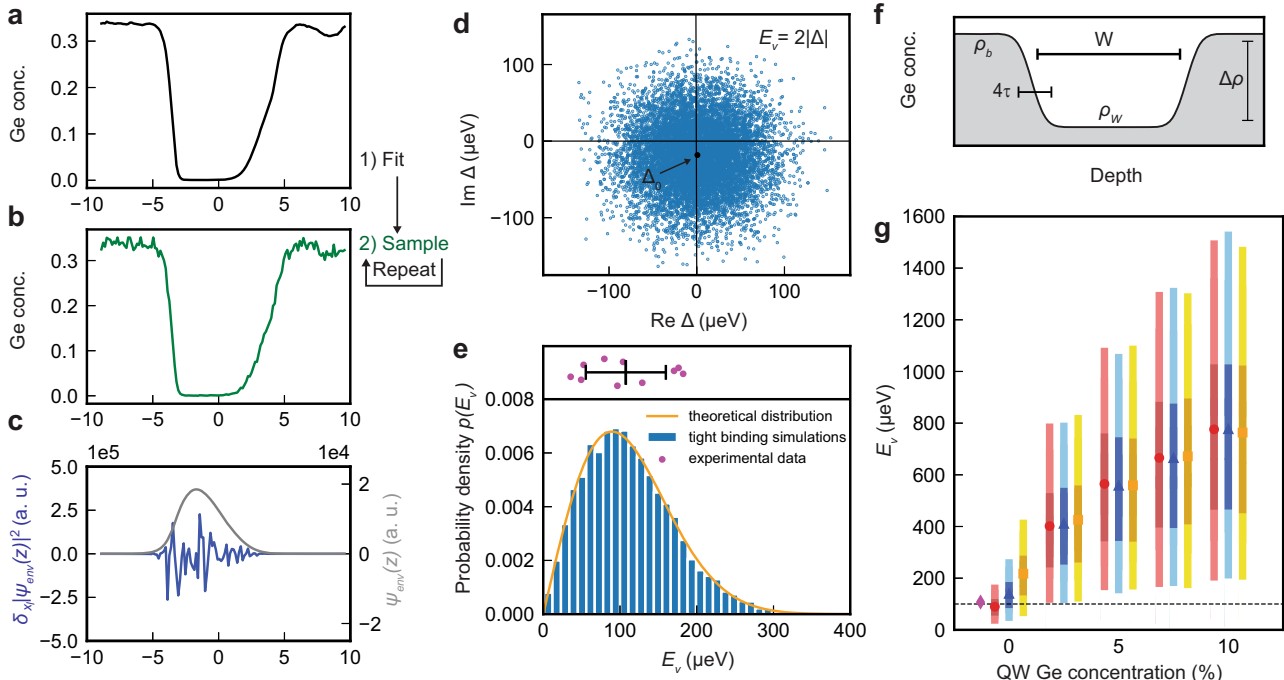

**Fig. 3 | Valley-splitting simulations. a** Average concentration profile obtained from APT data (quantum well A). **b** Typical, randomized Ge concentration profile, derived from **a**. **c** Envelope function $\psi_{env}(z)$, obtained for the randomized profile in **b** (grey curve), and the corresponding concentration fluctuations weighted by the envelope function squared: $\delta_{x_l}|\psi_{env}(z_l)|^2$ (blue). Here, the wavefunction is concentrated near the top interface where the concentration fluctuations are also large; the weighted fluctuations are therefore the largest in this regime. **d** Distribution of the intervalley matrix element $\Delta$ in the complex plane, as computed using an effective-mass approach, for 10,000 randomized concentration profiles. The black marker indicates the deterministic value of the matrix element $\Delta_0$, obtained for the experimental profile in **a**. **e** Histogram of the valley splittings from tight-binding simulations with 10,000 randomized profiles. The same profiles may be used to compute valley splittings using effective-mass methods; the orange curve shows a Rice distribution whose parameters are obtained from such

effective-mass calculations (see Methods). **f** Schematic Si/SiGe quantum well with Ge concentrations $\rho_W$ (in the well) and $\rho_b = \rho_W + \Delta\rho$ (in the barriers), with a fixed concentration difference of $\Delta\rho = 25\%$. **g** Distribution of valley splittings obtained from simulations with variable Ge concentrations, corresponding to $\rho_W$ ranging from 0 to 10%, and interface widths $4\tau = 5$ ML (red circles), 10 ML (blue triangles), or 20 ML (orange squares), where ML refers to atomic monolayers. Here, the marker describes the mean valley splitting, while the darker bars represent the 25-75 percentile range and the lighter bars represent the 5–95 percentile range. Each bar reflects 2000 randomized tight-binding simulations of a quantum well of width $W = 120$ ML. The magenta diamond at zero Ge concentration shows the average measured valley splitting of quantum well A. In all simulations reported here, we assume an electric field of $E = 0.0075$ V/nm and a parabolic single-electron quantum-dot confinement potential with orbital excitation energy $\hbar\omega = 4.18$ meV and corresponding dot radius $\sqrt{\hbar/m^*\omega}$.

Fig. 3c, we compute $\delta\Delta_l$ using the randomized density profile of Fig. 3b (blue curve). We see that most significant fluctuations occur near the top interface, where $|\psi_{env}(z_l)|$ and the Ge content of the quantum well are both large. In Fig. 3d we plot $\Delta$ values obtained for 10,000 quantum-well realizations using this effective mass approach. The deterministic contribution to the valley splitting $\Delta_0$ (black dot) is seen to be located near the center of the distribution in the complex plane, as expected. However, the vast majority of $\Delta$ values are much larger than $\Delta_0$, demonstrating that concentration fluctuations typically provide the dominant contribution to intervalley coupling.

The total valley splitting is closely related to the intervalley coupling via $E_v = 2|\Delta|$, and therefore exhibits the same statistical behavior. In Fig. 3e, the orange curve shows the Rice distribution whose parameters are derived from effective-mass calculations of the valley splitting (see Methods), using the same concentration profiles as the histogram data. The excellent agreement between these different approaches confirms the accuracy of our theoretical techniques (Supplementary Note 3d).

## Discussion

Based on the results obtained above, we now propose two related methods for achieving large valley splittings (on average), with high yields. Both methods are derived from the key insight of Fig. 3c: due to random-alloy fluctuations, the valley splitting is almost always enhanced when the electronic wavefunction overlaps with more Ge atoms. In the first method, we therefore propose to increase the width

of the interface ($4\tau$) as shown in Fig. 3f, since this enhances the wavefunction overlap with Ge atoms at the top of the quantum well. This approach is nonintuitive because it conflicts with the conventional deterministic approach of engineering sharp interfaces. The second method, also shown in Fig. 3f, involves intentionally introducing a low concentration of Ge inside the quantum well. The latter method is likely more robust because it can incorporate both deterministic enhancement of the valley splitting from a sharp interface, and fluctuation-enhanced valley splitting.

We test these predictions using simulations, as reported in Fig. 3g, where different colors represent different interface widths and the horizontal axis describes the addition of Ge to the quantum well. For no intentional Ge in the quantum well, as consistent with the heterostructure growth profile of our measured quantum dots, the calculations show a significant increase in the valley splitting with increasing interface width. Here, the narrowest interface appears most consistent with our experimental results (magenta marker), attesting to the sharp interfaces achieved in our devices. As the Ge concentration increases in the quantum well, this advantage is largely overwhelmed by concentration fluctuations throughout the well. A very substantial increase in valley splitting is observed for all concentration enhancements, even at the low, 5% level. Here, the light error bars represent 5–95 percentiles while dark bars represent 25–75 percentiles. At the 5% concentration level, our simulations indicate that >95% of devices should achieve valley splittings > 100 μeV. This value is more than an order of magnitude larger than the typical operation temperature of spin-

qubits and is predicted to yield a 99% readout fidelity[25]. This would represent a significant improvement in qubit yield for Si quantum dots. A recent report of SiGe quantum wells with oscillating Ge concentrations provides the first experimental evidence that intentionally placing Ge in the quantum well leads to significant variability and some of the highest recorded values of valley splitting[45].

In conclusion, we argue for the atomic-level origin of valley splitting distributions in realistic Si/SiGe quantum dots, providing key insights on the inherent variability of Si/SiGe qubits and thereby solving a longstanding problem facing their scaling. We relate 3D atom-by-atom measurements of the heterointerfaces to the statistical electrical characterisation of devices, and ultimately to underlying theoretical models. We observe qualitative and quantitative agreement between simulated valley splitting distributions and measurements from several quantum dots, supporting our theoretical framework. Crucially, we learn that atomic concentration fluctuations of the $^{28}$Si → Si-Ge heterointerface are enough to account for the valley splitting spread and that these fluctuations are largest when the envelope of the wavefunction overlaps with more Ge atoms. Moreover, while we have only incorporated random alloy disorder into our theoretical framework so far, we foresee that APT datasets including correlated disorder, such as steps, will be used to further refine our theoretical understanding of valley splitting statistics. Since atomic concentration fluctuations are always present in Si/SiGe devices due to the intrinsic random nature of the SiGe alloy, we propose to boost these fluctuations to achieve on average large valley splittings in realistic silicon quantum dots, as required for scaling the size of quantum processors.

Our proposed approaches are counter-intuitive yet very pragmatic. The interface broadening approach seems viable for hybrid qubits, which require valley splitting to be large enough to be usable but not so large as to be inaccessible. For single-electron spin qubits, which don't use the valley degree of freedom, the direct introduction of Ge in the quantum well appears better suited for targeting the largest possible valley splitting. By adding Ge to the Si quantum well in small concentrations we expect to achieve on average valley splitting in excess of 100 μeV. Early calculations from scattering theories[46] suggest that the added scattering from random alloy disorder will not be the limiting factor for mobility in current $^{28}$Si/SiGe heterostructures. However, an approximate two-fold reduction in electron mobility was recently reported when an oscillating Ge concentration of about 5% on average is incorporated in the Si quantum well[45]. We speculate that fine-tuning of the Ge concentration in the quantum well will be required for enhancing the average valley splitting while not compromising the low-disorder potential environment, which is important for scaling to large qubit systems. We believe that our results will inspire a new generation of Si/SiGe material stacks that rely on atomic-scale randomness of the SiGe as a new dimension for the heterostructure design.

## Methods

### Si/SiGe heterostructure growth

The $^{28}$Si/SiGe heterostructures are grown on a 100-mm n-type Si(001) substrate using an Epsilon 2000 (ASMI) reduced pressure chemical vapor deposition reactor equipped with a $^{28}$SiH$_4$ gas cylinder (1% dilution in H$_2$) for the growth of isotopically enriched $^{28}$Si. The $^{28}$SiH$_4$ gas was obtained by reducing $^{28}$SiF$_4$ with a residual $^{29}$Si concentration of 0.08%[47]. Starting from the Si substrate, the layer sequence for quantum well A comprises a 900 nm layer of Si$_{1-x}$Ge$_x$ graded linearly from $x = 0$ to 0.3, followed by a 300 nm Si$_{0.7}$Ge$_{0.3}$ strain-relaxed buffer, an 8 nm tensily strained $^{28}$Si quantum well, a 30 nm Si$_{0.7}$Ge$_{0.3}$ barrier, and a sacrificial Si cap. The layer sequence for quantum well B comprises a 1.4 μm step-graded Si$_{(1-x)}$Ge$_x$ layer with a final Ge concentration of $x = 0.3$ achieved in four grading steps ($x = 0.07$, 0.14, 0.21, and 0.3), followed by a 0.45 μm Si$_{0.7}$Ge$_{0.3}$ strain-relaxed buffer, an 8 nm

tensily strained $^{28}$Si quantum well, a 30 nm Si$_{0.7}$Ge$_{0.3}$ barrier, and a sacrificial Si cap. In quantum well A, the Si$_{0.7}$Ge$_{0.3}$ strain-relaxed buffer and the Si quantum well are grown at 750 °C without growth interruption. In quantum well B the Si$_{0.7}$Ge$_{0.3}$ strain-relaxed buffer below the quantum well is grown at a temperature of 625 °C, followed by growth interruption and quantum well growth at 750 °C. This modified temperature profile yields a sharper bottom interface for quantum well B as compared to quantum well A.

### Atom probe tomography

Samples for APT were prepared in a FEI Helios Nanolab 660 dual-beam scanning electron microscope using a gallium focused ion beam at 30, 16, and 5 kV and using a procedure described in detail in ref. 48. Before preparation, a 150–200 nm thick chromium capping layer was deposited on the sample via thermal evaporation to minimize the implantation of gallium ions into the sample. All APT analyses were started inside this chromium cap with the stack fully intact underneath. APT was carried out using a LEAP 5000XS tool from Cameca. The system is equipped with a laser to generate picosecond pulses at a wavelength of 355 nm. For the analysis, all samples were cooled to a temperature of 25 K. The experimental data are collected at a laser pulse rate of 200–500 kHz at a laser power of 8–10 pJ. APT data are reconstructed using IVAS 3.8.5a34 software and visualized using the AtomBlend addon to Blender 2.79b and Blender 2.92 software. For the Voronoi tessellation the reconstructed data sets were exported to Python 3.9.2 and then tessellated using the scipy.spatial.Voronoi class of SciPy 1.6.2. Note that in these analyses the interfaces are represented as an array of sigmoid functions generated perpendicular to the respective interface on 3 nm × 3 nm tiles that are 1 nm apart. This sacrifices lateral resolution to allow for statistical sampling of the elemental concentrations but preserves the atomic resolution along the depth axis that APT is known to provide upon constructing the interface as shown in Fig. 2a.

### Device fabrication

The fabrication process for Hall-bar shaped heterostructure field effect transistors (H-FETs) involves: reactive ion etching of mesa-trench to isolate the two-dimensional electron gas (2DEG); P-ion implantation and activation by rapid thermal annealing at 700 °C; atomic layer deposition of a 10-nm-thick Al$_2$O$_3$ gate oxide; deposition of thick dielectric pads to protect gate oxide during subsequent wire bonding step; sputtering of Al gates; electron beam evaporation of Ti:Pt to create ohmic contacts to the 2DEG via doped areas. All patterning is done by optical lithography. Quantum dot devices are fabricated on wafer coupons from the same H-FET fabrication run and share the process steps listed above. Double-quantum dot devices feature a single layer gate metallization and further require electron beam lithography, evaporation of Al (27 nm) or Ti:Pd (3:27 nm) thin film metal gates, and lift-off. For linear quantum dot arrays the gate stack consists of 3 layers of Ti:Pd metallic gates (3:17, 3:27, 3:27 nm) isolated from each other by 5 nm Al$_2$O$_3$ dielectric interlayers. The fabrication processes for quantum dot devices are further detailed in ref. 49.

### Electrical characterisation of devices

Hall-bar measurements are performed in a Leiden cryogenic dilution refrigerator with a mixing chamber base temperature $T_{MC} = 50$ mK[50]. We apply a source-drain bias of 100 μV and measure the source-drain current $I_{SD}$, the longitudinal voltage $V_{xx}$, and the transverse Hall voltage $V_{xy}$ as function of the top gate voltage $V_g$ and the external perpendicular magnetic field $B$. From here we calculate the longitudinal resistivity $\rho_{xx}$ and transverse Hall resistivity $\rho_{xy}$. The Hall electron density $n$ is obtained from the linear relationship $\rho_{xy} = B/en$ at low magnetic fields. The carrier mobility $\mu$ is extracted from the relationship $\sigma_{xx} = ne\mu$, where $e$ is the electron charge. The percolation density $n_p$ is extracted by fitting the longitudinal conductivity $\sigma_{xx}$ to the relation

$\sigma_{xx} \propto (n - n_p)^{1.31}$. Here $\sigma_{xx}$ is obtained via tensor inversion of $\rho_{xx}$ at $B = 0$. Quantum dot measurements are performed in Oxford and Leiden cryogenic refrigerators with base temperatures ranging from 10 to 50 mK. Quantum dot devices are operated in the few-electron regime. Further details of the 2DEG and quantum dot measurements are provided in the Supplementary Note 1.

## Theory and simulations

The quantum-well potential at vertical position $z_l$ is simply defined here as a linear interpolation of the conduction-band offset at the quantum-well interface: $U(z_l) = \frac{x_l^d - x_s}{x_w - x_s} \Delta E_c$, where $x_l^d$ is the average Si concentration in layer $l$, $x_s$ is the average Si concentration in the strain-relaxed SiGe barriers, $x_w$ is the average Si concentration in the strained quantum well, and $\Delta E_c$ is the conduction band offset in the absence of fluctuations. In the effective-mass theory, the intervalley coupling matrix element can then be approximated by the sum

$$\Delta = \frac{a_0}{4} \sum_l e^{-2ik_0 z_l} \frac{x_l^d - x_s}{x_w - x_s} \Delta E_c |\psi_{\text{env}}(z_l)|^2. \tag{1}$$

Defining the local concentration fluctuations as $x_l^d = \bar{x}_l + \delta_{x_l}$, the matrix element can then be split into its deterministic and fluctuating contributions $\Delta = \Delta_0 + \delta\Delta$, where the fluctuating term $\delta\Delta$ contains all dependence on $\delta_{x_l}$:

$$\delta\Delta = \frac{a_0}{4} \frac{\Delta E_c}{x_w - x_s} \sum_l e^{-2ik_0 z_l} \delta_{x_l} |\psi_{\text{env}}(z_l)|^2. \tag{2}$$

The deterministic term $\Delta_0$ represents the matrix element of the ideal, smooth concentration profile, while $\delta\Delta$ describes the fluctuations about this value. For concentration fluctuations $\delta_{x_l}$ defined by binomial distributions of Ge and Si atoms, the resulting valley splitting $E_v = 2|\Delta_0 + \delta\Delta|$ corresponds to a Rice distribution with parameters $v = 2|\Delta_0|$ and $\sigma = \sqrt{2}\sqrt{\text{Var}[\delta\Delta]}$[51]. For additional details, see Supplementary Note 3. All simulations and numerical calculations reported in this work were performed using Python 3.7.10 with the open-source libraries NumPy, SciPy, and Matplotlib. The 3D atomistic simulations were done using the large-scale Slater-Koster tight-binding solver NEMO3D. A spin-resolved 20 band sp3d5s* nearest neighbour model was used. Strain optimization was done using a valence force field Keating model.

## Data availability

All data included in this work are available from the 4TU. Research Data international data repository at https://doi.org/10.4121/16592522.

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

## Acknowledgements

This work was supported in part by the Army Research Office (Grant No. W911NF-17-1-0274). The views and conclusions contained in this document are those of the authors and should not be interpreted as representing the official policies, either expressed or implied, of the Army Research Office (ARO), or the U.S. Government. The U.S. Government is authorized to reproduce and distribute reprints for Government purposes notwithstanding any copyright notation herein. The APT work was supported by NSERC Canada (Discovery, SPG, and CRD Grants), Canada Research Chairs, Canada Foundation for Innovation, Mitacs, PRIMA Québec, and Defence Canada (Innovation for Defence Excellence and Security, IDEaS).

## Author contributions

A.S. grew and designed the Si/SiGe heterostructures with B.P.W. and G.S. M.P.L, R.R., S.N.C, and M.F. developed the theory. S.K performed atom probe tomography and analyzed the data with B.P.W., and L.A.E.S. M.L. fabricated heterostructure field effect transistors measured by B.P.W. S.A. and N.S. fabricated quantum dot devices measured by B.P.W., A.J.Z., S.G.J.P., M.T.M., X.X., G.Z, and N.S. S.N.C., O.M., M.F., and G.S. supervised the project. G.S. conceived the project. B.P.W, M.P.L, S.K., and G.S. wrote the manuscript with input from all authors.

## Competing interests

M.P.L., S.N.C., and M.F. declare a related patent application that proposes to intentionally introduce Ge into the quantum well, to increase the average valley splitting: US Patent Application No. 63/214957 (currently under review). The remaining authors declare no competing interests.
