## [Peer Review File · Nature Communications]

Reviewer 1 (Remarks to the Author(s))

The Authors explore the field of controlling the valley splitting in silicon quantum dots, which is a major issue currently affecting the potential scaling of silicon qubits despite the natural fit of silicon with large scale production. The Authors experimentally analyse the valley splitting of a reasonable number of devices from two distinct quantum wells and connect such results with the experimental observation by atom probe tomography so to obtain atomic scale distribution of the Ge close to the interface with Si28. The experiments are supported by simulation which explains the connection between the valley splitting spectroscopy and the APT. They conclude that increasing the width of the interface "on average" (line 360) increases the valley splitting. Second, they run simulations where randomly diffused Ge is added in low concentration in the Si28 layer, which, according to the random fluctuations as from those observed by the APT, the valley splitting is statistically enhanced.

Overall, the manuscript involves an impressive amount of work with several experimental techniques and is supported by a detailed understanding based on robust simulations. Unfortunately, if seen from the perspective of the application to scaling to (hopefully millions of) silicon qubits this statistical method looks not convincing to engineer the control of the valleytronics of individual electrons, which is the issue that the Authors would like to address and, if addressed, would make the article suitable for this journal. To compare with existing literature, I have in mind the research appeared simultaneously to the preprint of this manuscript on the arxiv: 2112.09765 McJunkin et al. "SiGe quantum wells with oscillating Ge concentrations for quantum dot qubits" which involves some Authors which are also Authors of this manuscript, where there is 1) an engineered control by a modulated concentration of the Ge 2) an experimental verification of the simulations to show that the postulated method actually works.

Unfortunately, I see the two following reasons preventing this manuscript to appear on Nature Communication, namely the not-predictable effect on each quantum dot of the suggested methods, and the lack of an experimental demonstration that the recipe of low doping of Ge in the Si actually works (and with which yield, with a complete assessment of the method when put in practice).

To have the expected impact in the field required by Nature Communications and therefore to be published, in my opinion the Authors should indicate also how to reduce/remove the variability of the resulting valley splitting increased in average, and to experimentally support the validity of their method. Said this, I believe that this manuscript is of interest for the silicon quantum dot community and it should be published on a different, more specific, journal. I appreciate and understand the value of the huge work done, so I'm very sorry I can not be more positive this time. Enrico Prati

Our reply:

We thank Reviewer 1 for bringing his viewpoint forward and openly (to the point of disclosing his identity). We feel especially rewarded for the recognition of the *value of the huge work* and for acknowledging that our study *involves an impressive amount of work with several experimental techniques and is supported by a detailed understanding based on robust simulations*. In the Reviewers report we dont see any remarks or objections regarding what we consider the core results of the manuscript: to solve the outstanding challenge of linking the valley splitting in Si/SiGe quantum dots to the relevant atomic-scale material properties, that hinders the development of accurate and predictive theoretical models, and establish comprehensive insights into the atomic-level origin of valley splitting in realistic silicon quantum dots. This is the main claim of the paper and it was spelled out in the introduction of the paper.

The criticism from Reviewer 1 is, instead, pointing only to the last section of the paper. In that section, we use our accurate and predictive model to propose on one possible practical strategy for mitigating the valley splitting issue in Si/SiGe. Our crucial learning is about the critical role of random fluctuations, which is inherent to the SiGe alloy, in relationship to valley splitting. The variability of valley splitting is unavoidable

in any devices containing SiGe. We are just suggesting a method that takes this into account and maximizes success rates for scaling up to many qubits. Let us be absolutely clear: in the context of this study, our proposition is intended as a forward looking section, supported by simulations. It occupies, in the narrative of the paper, the two final paragraphs before the conclusions and one figure panel (Fig 3g). In our view, pursuing experimentally this strategy with the rigor and statistical relevance it deserves is a completely different endeavor that goes well beyond the scope of this paper. We note that on the topic of valley splitting in Si/SiGe, other structures have been proposed but not put in practice within the same publication in Nature Communications (for example Zhang et al, Genetic design of enhanced valley splitting towards a spin qubit in silicon, Nature Communication 4,2396 (2013)]. In our case, we complement our core experimental section with a strong theoretical framework that leads us to a proposed structure. Furthermore, results from the companion back-to-back paper arxiv:2112.0976 show that indeed including Ge in the Si quantum well brings improvement in valley splitting and prove our claims about variability, corroborating our proposition.

In any case, one may find our proposed solution defensible or opposable. We acknowledge and respect that Reviewer 1 dissents from our propositions. However, the key point is that, within the limits of the theoretical framework and current understanding, our simulations are technically correct. Therefore, we politely express our concern that our work is rejected based on a debatable opinion about an outlook proposition. To the contrary, we find it remarkable if this section fuels dialogue and discussion in the community (with some scholars supporting and other opposing) and it is a very good sign that this paper is already having an impact even before it has been published. Reminding ourselves that Nature Communication aims to *publish high-quality research in all areas of physical science* and that *Papers published by the journal aim to represent important advances of significance to specialists within each field* we strongly invite Reviewer 1 to reconsider his first judgment.

We do recognize, however, that our manuscript needed further editing and very careful rewording to make sure that the main claims (experimentally/theoretically) are spelled out clearly and that our proposed strategy is correctly perceived by the reader as a strong outlook, which already has seen first experimental confirmations in the manuscript arxiv:2112.0976 .

The following changes in the manuscript go in this direction:

Abstract:

Here, we elucidate, predict, and control the valley splitting by the holistic integration of 3D atomic-level properties, theory and transport.

changed into:

*Here, we **elucidate and predict** the valley splitting by the holistic integration of 3D atomic-level properties, theory and transport*

End of the introduction section:

Based on these atomistic insights, we propose a practical strategy to statistically enhance valley splitting

changed into:

*Based on these atomistic insights, we **conclude by proposing** a practical strategy to statistically enhance valley splitting*

The concluding note *We are aware of related work in which Ge is intentionally placed in the quantum well to affect valley splitting [mcjunkin_sige_2021]*

was deleted and rephrased as following in the Discussion section:

A recent report of SiGe quantum wells with oscillating Ge concentrations provides the first experimental evidence that intentionally placing Ge in the quantum well leads to significant variability and some of the highest recorded values of valley splitting[11].

In addition to these editing, Section Headings have been added and, importantly, our proposition of adding Ge to the Si quantum well for increasing on average valley splitting falls now into the **Discussion Section** as a hypothesized pathway that we are introducing based on the Results section.

Reviewer 2 (Remarks to the Author(s))

The manuscript by Wuetz et al describes a study of valley splitting in Si/SiGe quantum dots that incorporates sophisticated atomic-level characterization of the quantum well structure in addition to magnetospectroscopy to claim that the statistical variation in measured valley splittings is a consequence of random alloy fluctuations. Further, they propose two concept structures based on this finding they purport will statistically improve valley splitting in actual devices. This work is interesting and timely given all of the very recent progress in Si-based spin qubit performance (as evidenced by several high impact publications in Nature and elsewhere) and the well-known challenge that valley splitting presents to scaling up these systems to computation relevance.

The manuscript is broken up into three sections.

The first describes the devices and the magnetospectroscopy technique used to infer valley splitting. Here we arrive at my first issue, which is the unsubstantiated claim that the two electron singlet-triplet splitting measured using magnetospectroscopy is the same as the valley splitting. It is widely known that orbital effects, especially from electrons confined in asymmetric quantum dots, can reduce the singlet-triplet splitting below the valley splitting. That authors state that due to the small size and strong confinement in our quantum dots, we expect exchange corrections to have negligible effects: here we take $E_v \approx E_{st}$. Given how important these measurements are to the later claims, I would require further substantiation of this claim. For example, what are the measured orbital energies? Also, there are well-known techniques for extracting the valley splitting unambiguously from single electron measurements (pulsed gate spectroscopy, photon assisted tunneling, detuning axis pulsed spectroscopy, etc) so it seems odd to me to pin all of the supporting data on this unsubstantiated claim.

The second major section describes APT characterization of the quantum well. While APT has been used to characterize Si/SiGe quantum wells previously (Dyck et al, Adv Mater Interfaces 2017, 1700622) this work was not cited. They describe in great detail their analysis method for obtaining interface information from APT, but it is very unclear how it tells a different story than the HAADF-STEM images shown previously. The closest thing I could find to a conclusion is that the RMS roughness for an iso-concentration surface was consistent with that expected from uncorrelated random alloy fluctuations, which means that the Ge atoms do not form clumps but really appear in the lattice in a completely random way. It may even be possible to infer that same information from the HAADF-STEM image, but this is not clear. The authors also do not state what advantages APT has over HAADF-STEM in terms of interface determination. Given the amount of space dedicated to this section, it should be better tied to the main conclusions.

The final section describes theoretical modeling of the valley splitting resulting from alloy fluctuations. The authors take the average 1D Ge concentration profile obtained from APT (which is equivalent to the HAADF-STEM image) and then create individual instantiations of that profile given the random alloy fluctuations and the typical size scale of a trapped electrons wavefunction. They calculate the resulting valley splitting (the imaginary mixing matrix element) and repeat that for several thousand alloy instantiations. From this they determine a distribution of valley splittings and qualitatively compare it to the measurements from the first part. They find a broad distribution of valley splitting and note that the distribution is completely dominated by the alloy fluctuations: the averaged well structure only provides a minute offset in the complex map of the coupling matrix element. Based on this finding, they propose a shift from the conventional approach of engineering valley splitting through interface sharpening and instead aim to improve it statistically through introducing more interaction with Ge alloy fluctuations. In particular they propose two paths: increasing the width of the interface and/or putting a uniform concentration of Ge in the well itself. While they discuss the first option like its a real option, subsequent analysis in figure 3g shows that the width widening approach is a small effect only useful when there is no Ge in the well itself. Once Ge is introduced into the well, the valley splitting distribution only depends weakly on the interface width. The authors conclude by saying that such an approach would result in a significant improvement in qubit yield for Si quantum dots but this is a hard claim to justify since it is unclear what role valley splitting plays in qubit yield (like how does qubit performance or yield depend on valley splitting? Does it play a role aside from initialization and measurement?). They also

speculate without substantiation that the introduction of Ge into the well would not limit mobility or otherwise impact the low-disorder potential needed for spin qubits.

Overall I found this manuscript to have several interesting elements but they don't tie together very well and many claims are dubious or wholly unsubstantiated. Without significant redress, I cannot recommend publication at this time. As a summary, my main concerns are the following:

Our reply: We thank Reviewer 2 for the thorough and detailed review of our manuscript, which is reflected and comes together in the extensive report received. The point-by-point summary is also very helpful to organize our review effort. It is thanks to constructive reviews that progress can be made, in general, and in particular for this complex manuscript, where we aim to combine different elements in a holistic approach to valley splitting. We agree with the general sentiment of the Reviewer that work is needed to tie things together better, and we clarify/substantiate accordingly our claims in the point-by-point reply below and in the manuscript.

2.1 The use of measured singlet-triplet splittings as a measure of valley splitting is generally not a wise approach. There are means to measure the valley splitting directly, but at a minimum the authors should provide additional measurements of the orbital energies and calculations to show that indeed the singlet-triplet splittings are a measure of the valley splitting.

2.2 The authors several times refer to an assumption about the orbital energy. Why can't this just be measured?

We thank the reviewer for bringing up these two comments about valley splitting, singlet-triplet splitting, and orbital energies. Indeed, we measure singlet-triplet splitting in magnetospectroscopy and assume that these singlet-triplet splittings are representative of valley splitting, given the large orbital energy in our strongly confined dots. In Si/SiGe heterostructures, large orbital splittings on the order of meV have often been reported [7, 26, 13] with dot dimensions similar to ours. The measured valley splittings in these reports are typically one order of magnitude smaller than the orbital energy. To substantiate better our assumptions, as asked by Reviewer 2, we also include now in Supplementary Fig. 3 a measurement of the orbital splitting for a specific device, which was very similar to all the devices studied in this work. We first estimate the lever arm using Coulomb diamonds and extract $\alpha = 0.11 \text{ eV/V}$. We then perform pulsed gate spectroscopy where we find an orbital energy of $\alpha \times \delta V_{orb} = 4.18 \text{ meV}$. This value is more than an order of magnitude larger than our typical singlet-triplet splitting and indicates strong confinement in our dots. For strongly confined dots, as explained in H. Ercan et. al, Phys. Rev. B 104, 235302 (2021), the singlet-triplet splitting is nearly equal to the valley splitting.

In addition to the new Supplementary Fig. 3, we modified the following sentence in the section about valley splitting measurement:

E_{ST} sets a lower bound on the valley splitting, $E_v \geq E_{ST}$ [2, 5]. Due to the small size and strong confinement in our quantum dots, we expect exchange corrections to have negligible effects; here we take $E_v \approx E_{ST}$.

into:

E_{ST} sets a lower bound on the valley splitting, $E_v \geq E_{ST}$ [2, 5]. Due to small size, our dots are strongly confined with orbital energy much larger than E_{ST} (Supplementary Fig. 3), similar to other Si/SiGe quantum dots [26, 13, 8]. Therefore, we expect exchange corrections to have negligible effects [5] and here we take $E_v \approx E_{ST}$.

2.3 The authors should clarify what exactly the value of the APT characterization is over HAADF-STEM. Is it just a verification of non-clustering of Ge atoms or am I missing something? There is a lot of text dedicated to this analysis but it's unclear if it's necessary for the rest of the story to be true.

Our reply: As spelled out in the abstract and introduction, APT provides a unique insight into the 3D landscape of the Si/SiGe interface that is impossible to achieve with any other technique. Yes, one critical

insight we gain in our paper is the non-clustering of GeSi in most of the samples analysed. As mentioned in the main text, and detailed in the Supplementary section 2e and Supplementary Fig. 13, there are also a few instances where we observe features that are compatible with correlated disorder from atomic steps, which is a significant first for Si/SiGe heterostructures[4].

There are other advantages of APT. For example, APT brings added value compared to HAADF-STEM because:

- a) the presence or absence of any kind of pattern would not be resolved by HAADF-STEM due to the 2D images and
- b) it provides the seeding for the theoretical model.

Concerning the first point, in principle all kinds of issues could cause a blurring of the interface in HAADF-STEM - for example if the interface is

- tilted with respect to the TEM viewing direction
- wavy on a scale much larger than the TEM lamella thickness
- rough on a scale smaller than the TEM lamella thickness
- a series of atomic steps in the direction perpendicular to the TEM lamella

While some of these scenarios are unlikely due to the symmetries inherent to this material system, APT just straight out proves that they are all non-existent.

Concerning the second point, we used the data from APT to seed the theoretical models. In particular, we used the Ge concentrations and Ge fluctuations observed in APT to randomly generate interfaces for the 1D model. Once it is known that the in-plane Ge distribution is random, and this is critical knowledge we gain only by APT, the fluctuations could have been predicted from any 1D elemental profile with sufficient (=atomic) depth resolution but getting elemental distributions from HAADF-STEM image contrast is not straightforward at all (see for example [15]) nor is it nearly as accurate as APT - so it cannot easily be used to seed the Ge concentrations either. In other words, we expect the interface width in APT and HAADF-STEM to be approximately the same but we don't expect the interface 'shape' (=the 1D fit function) to be the same. Furthermore, since we did observe in some instances features compatible with atomic steps, it is interesting to expand our theory by seeding it with these more complex configurations of Ge/Si atoms, to assess the relevance or not of atomic steps to valley splitting, a long standing open question in the field. This will be the subject of a forthcoming theory manuscript, as explained in point 3.7. Results shown in Figure R1 of this report, point to alloy disorder as the predominant factor in determining valley splitting in realistic samples with realistic (broadened) interfaces.

In the revised manuscript we have changed the following sentences in the APT section to clarify and better highlight the significance of our characterisation:

Sentence expanded in the second paragraph of the APT section

To probe the interface properties with the highest possible resolution allowed by APT and differently from previous APT studies on Si/SiGe[4], we represent the atom positions in the acquired data sets in form of a Voronoi tessellation[23, 22] and generate profiles on an $x - y$ grid of the tessellated data, as described in Supplementary Discussion Section 2c.

Sentence added at the end of second paragraph:

Importantly, Ge concentration isosurfaces as shown in Fig. 2b,c are constructed by determining the vertical position for which each of the sigmoids reaches a specific concentration. Note, that we oversample the interface to improve the lateral resolution by making the $3 \text{ nm} \times 3 \text{ nm}$ tiles partially overlap (Supplementary Discussion Section 2c).

Sentence expanded in the fourth paragraph:

To pinpoint the root cause of atomic-scale fluctuations at the interface, in Fig. 2f,g we *utilize the 3D nature of the APT data sets, calculate, and compare the root mean square (RMS) roughness of the interfaces (solid green lines) as measured by APT on quantum well B to two 3D models (Fig. 2f,g) mimicking the dimensions of an APT data set.*

Finally, we have added four Supplementary Movies for both top bottom interface within quantum well A and B, showing the evolution across the of each isoconcentration surfaces from 1-30 % Ge. Obviously, all of this is impossible to achieve with HAADF-STEM. We changed the caption of Supplementary Fig. 10 accordingly:

Example of Germanium isoconcentration surfaces on the top (a, b) and bottom (c, d) interfaces of both Quantum Wells a and B. The plots reported here show one particular isosurface, 1% in a, b and 30% in c,d. Animated short clips provided as Supplementary Movies show the evolution across the interfaces of each isoconcentration surfaces, from 1-30 % Ge. As before the depth for each map can be extracted from the sigmoid fits to the profile in each cell.

- 2.4 **Given the limited statistics of valley splitting data, its tough to say anything more than a qualitative statement that alloy fluctuations can give rise to distributions broad enough to match the data. In and of itself it is not proof that the measured distribution does result from this type of physics, only that it could. It would be far more satisfying if a more definitive statement could be made.**

Our reply:

We thank Reviewer 2 for this comment. We still highlight and emphasize that our study is the first of its kind to actually compare statistical results of valley splitting with simulated valley splitting distribution. This was also recognized by Reviewer 3, stating that *we report on magnetospectroscopy measurements of the valley splitting of 22 Si/SiGe quantum dots from 2 wafers and that Such large statistics is unprecedented and required to gain understanding of the variation.* However, we agree that, strictly speaking, we demonstrate compatibility rather than a definitive proof that measured valley splitting distribution result from alloy disorder. Further simulations provided in Fig. R1 and R2 provide further arguments about the dominant effect of alloy scattering on valley splitting. We have carefully rephrased the wordings of our claims accordingly.

In the abstract, we modify the sentence

We find that the concentration fluctuations of Si and Ge atoms within the 3D landscape of Si/SiGe interfaces explain the observed large spread of valley splitting from measurements on many quantum dot devices.

into:

*We find that the concentration fluctuations of Si and Ge atoms within the 3D landscape of Si/SiGe interfaces **can** explain the observed large spread of valley splitting from measurements on many quantum dot devices.*

In the introduction, we change the sentence

By comparing theory with experiments, we find that the measured random distribution of Si and Ge atoms at the Si/SiGe interface determines the measured valley splitting spread in real quantum dots.

into:

*By comparing theory with experiments, we find that the measured random distribution of Si and Ge atoms at the Si/SiGe interface **is enough to account for** the measured valley splitting spread in real quantum dots.*

In the conclusion, we modify the sentence

In conclusion, we establish the atomic-level origin of valley splitting distributions in realistic Si/SiGe quantum dots, providing key insights on the inherent variability of Si/SiGe qubits and thereby solving a long-standing problem facing their scaling.

into:

*In conclusion, we **argue for** the atomic-level origin of valley splitting distributions in realistic Si/SiGe quantum dots, providing key insights on the inherent variability of Si/SiGe qubits and thereby solving a longstanding problem facing their scaling.*

Also in the conclusion, we modify the sentence

Crucially, we learn that atomic concentration fluctuations of the $^{28}\text{Si} \rightarrow \text{Si-Ge}$ heterointerface are the driving force for the valley splitting spread and that these fluctuations are largest when the envelope of the wavefunction overlaps with more Ge atoms.

into:

*Crucially, we learn that atomic concentration fluctuations of the $^{28}\text{Si} \rightarrow \text{Si-Ge}$ heterointerface are **enough to account** for the valley splitting spread and that these fluctuations are largest when the envelope of the wavefunction overlaps with more Ge atoms.*

- 2.5 The authors should discuss real tradeoffs between the interface widening approach and the direct Ge introduction into the well. The latter could clearly have a much strong benefit to valley splitting, but I would imagine this could introduce other problems like significant mobility degradation and subsequent poor disorder performance. As it is presented now, its unclear why one would pursue the interface widening approach. I would strongly advocate for a clear and quantitative discussion of tradeoffs.**

Our reply:

We thank Reviewer 2 for this comment. Indeed, we agree about the importance of a quantitative discussion of the advantages and disadvantages of broadening the interface versus adding Ge to the quantum well. As discussed in the response to Reviewer 1, these proposals to increase E_v are an outlook for future work and not the primary focus of the paper. However, such a discussion of the differences between these approaches will be included in a future publication. As a comment, some qubit designs, like the hybrid qubit, utilize the valley degree of freedom. For these qubits, we don't want very small E_v , but we also don't need massive E_v either. In this case, it might make sense to use the interface broadening approach. It will boost E_v so it is generally large enough to be usable, but not so large as to be inaccessible. For devices which don't use the valley degree of freedom, like the single-electron spin qubit, we would want the valley splitting to be large enough so excited valleys don't contribute. In these cases, it makes much more sense to add Ge to the well, which will boost E_v much more.

A quantitative analysis is beyond the scope of this work and it will be the subject of a forthcoming theoretical publication, but in the Discussion section, we now added the following sentence to provide our viewpoint and stimulate discussions in the community:

Sentence added in the concluding paragraph:

The interface broadening approach seems viable for hybrid qubits, which require valley splitting to be large enough to be usable but not so large as to be inaccessible. For single-electron spin qubits, which dont use the valley degree of freedom, the direct introduction of Ge in the quantum well appears better suited for targeting the largest possible valley splitting.

The impact of alloy scattering on mobility was mentioned based on the theoretical calculations in Ref.[14]. Also in this case, experimental studies to understand with statistical significance the scattering properties of SiGe quantum wells with small Ge concentrations are necessary, as a stepping stone towards quantum dots.

- 2.6 The language describing the potential impact to qubit yield is far too vague. The authors do not even posit how valley splitting impacts qubit yield, or what values it needs to take on**

in order to be good enough. Without that background it is very difficult to understand how important this work is.

Our reply:

We thank Reviewer 2 for this comment. The most intuitive way to understand how valley splitting impacts qubit yield is to simply consider the qubit energy manifold in relation to the electron temperature of qubits. For a silicon quantum dot to behave as an ideal single electron spin 1/2 qubit, valley splitting should be as high as possible, as the qubit can no longer be properly defined if higher valley states are occupied. Essentially, the presence of the additional valley leads to non-computational states into which quantum information can leak. This is even more crucial as we scale the number of qubits, since the number leakage states will scale exponentially with the number of electrons.

In the outlook section of the manuscript, where we propose alternative stacks for a statistical boost of valley splitting, we arbitrarily chose 100 μeV ($\simeq 1.1$ K) as a reasonable and "good enough" value for increasing the qubit yield for a number of reasons:

- This chosen value is more than one order of magnitude larger than the typical operation temperature of spin qubits (≤ 100 mK), implying that higher valley population is essentially suppressed.
- Theoretical studies have shown that with valley splitting of 100 μeV , readout fidelity in excess of the fault-tolerant threshold of 99% may also be achieved[20], which is a key requirement to perform high fidelity quantum operation.
- In recent qubit experiments, Si/SiGe quantum dot systems with valley splitting in excess of 100 μeV have indeed shown 100% qubit yield in a Si/SiGe linear array of six quantum dots[16] and, in a two-qubit system, two-qubit gate fidelity above 99%, enabling computing with spin qubits at the surface code error threshold[25].

All these considerations substantiate 100 μeV as a reasonable target to aim for when scaling spin-qubits to larger systems in the near term.

To address the Referee's comment, we reworked and expanded the introduction section, with references that were overlooked and with a more rigorous language to explain the significance of valley splitting to qubit operation, with the following text

However, spin-qubits in silicon suffer from a two-fold degeneracy of the conduction band minima (valleys) that creates several non-computational states that act as leakage channels for quantum information[27]. These leakage channels increase exponentially with the qubit count[17], complicating qubit operation and inducing errors during spin transfers. Despite attempts to enhance the valley energy splitting, the resulting valley splittings are modest in Si/SiGe heterostructures, with typical values in the range of 20 to 100 μeV [24, 26, 19, 18, 6, 13, 1, 12] and only in a few instances in the range of 100 to 300 μeV [2, 8, 3]. Such variability in realistic silicon quantum dots remains an open challenge for scaling to large qubit systems. In particular, the probability of thermally occupying the excited valley state presents a challenge for spin initialization, and, in some cases, intervalley scattering may limit the spin coherence[9]. Furthermore, small valley splitting may affect Pauli spin blockade readout[20], which is considered in large-scale quantum computing proposals[21, 10]. Therefore, scaling up to larger systems of single-electron spin qubits requires that the valley splitting of all qubits in the system should be much larger than the typical operation temperatures (20 – 100 mK).

Outlook Section, we now explain why 100 μeV is a good target value:

A very substantial increase in valley splitting is observed for all concentration enhancements, even at the low, 5% level. Here, the light error bars represent 5-95 percentiles while dark bars represent 25-75 percentiles. At the 5% concentration level, our simulations indicate that >95% of devices should achieve valley

splittings $> 100 \mu\text{eV}$. This value is more than an order of magnitude larger than the typical operation temperature of spin-qubits and is predicted to yield a 99% readout fidelity[20]. This would represent a significant improvement in qubit yield for Si quantum dots. A recent report of SiGe quantum wells with oscillating Ge concentrations provides the first experimental evidence that intentionally placing Ge in the quantum well leads to significant variability and some of the highest recorded values of valley splitting[11].

Reviewer 3 (Remarks to the Author(s))

The manuscript atomic fluctuations lifting the energy degeneracy in Si/SiGe quantum dots by B. P. Wuetz, M. P. Losert and S. Koelling et al. combines three scientific studies, in order to improve the understanding of the variation of the valley splitting among quantum dots, which is a very timely and significant topic for Si/SiGe qubits. Electron spin qubits in Si/SiGe are the furthest developed, but suffer from uncontrolled low valley splitting. Increasing the valley splitting for large arrays of quantum dots is one of the remaining challenges of scalable quantum computation in Si/SiGe.

First, the authors report on magneto-spectroscopy measurements of the valley splitting of 22 Si/SiGe quantum dots from 2 wafers by. Such large statistics is unprecedented and required to gain understanding of the variations.

Second, the authors analysed 10 samples from two SiGe wafers by atomic probe tomography to measure the Ge concentration profile in 3D at the Si/SiGe interface. Remarkably, they introduce a new method to achieve monolayer resolution in the growth direction and 3 nm lateral resolution. Such high resolution is required to model the valley splitting, which depends on the very details of the Si/SiGe interfaces of the quantum well.

Third, the authors simulate the valley splitting using interface width measured from APT and expected alloy disorder as an input parameter. They use tight-binding model in 1D and partly in 3D and effective mass theory. Most importantly, the authors combine all three scientific studies and thus can connect material characterisation by simulation to the measured valley splitting. This connection is significant, unprecedented and contains some innovations within the material analysis and theory. The used statistics is important to support the authors claim that the measured variations of the valley splitting matches the distribution expected from theory. Finally, they used the theory to predict a method to increase the mean value of the valley splitting.

The presentation of the measurements and the methods is transparent and clear due to the supplementary information. The broad range of studies/methods and its relation to the timely and significant issue of scalability of Si/SiGe qubits justifies publication in Nature Communication after some issues has to be addressed/clarified. The authors made impressive progress on the issue of valley splitting, but the manuscript could improve by clearly stating weaknesses within the studies and open questions.

3.1 The authors measure singlet-triplet splitting and set this equal to valley splitting in the quantum dot. Actually, the singlet-triplet splitting is a lower bound for valley splitting and its relation depend on the quantum dot size. The quantum dot size varies in this study, since quantum dots are formed with various gate patterns. Fig. 1d is only one example of the used gate patterns. The authors should be more clear about this in the main text.

Our reply: We thank the reviewer for this and for other supporting comments. All singlet-triplet-splitting measurements are performed on quantum dots with only two different designs, i.e. single-layer and multi-layer quantum dots. Within these two designs, plunger gate diameters vary between 40-50 nm, where the confinement of the quantum dots should not significantly change. For clarity, we added the range of the plunger gate diameter $d_p = 40 - 50$ nm in the main text and the diameter of each quantum dot in the Supplementary Table 1. Furthermore, we have added for one quantum dot (Supplementary Fig. 3) the measurements of the orbital energy, confirming that is more than one order of magnitude higher than the typical singlet-triplet splitting, substantiating our assumptions that singlet-triplet splitting approximates valley splitting (see reply to point 2.1 above).

3.2 The authors measure one valley spitting on wafer B as exactly 0. The related raw data shows no dependence on B-field although a negative slope is expected for T- (Fig. S5e). The measurement might be wrong. At least the valley splitting should be given as an upper bound or with a proper error.

Our reply:

We thank Reviewer 3 for this comment about our report of "zero" valley splitting. Let us clarify better what we mean with zero valley splitting. We deem the magnetospectroscopy data to be correct. Looking at the energy diagram schematics in Supplementary Fig 4b we see that at magnetic fields $B > B_{ST}$ the singlet-state S_0 becomes occupied and we expect a straight line in our magnetospectroscopy experiment parallel to the x-axis (the B-field axis). If $E_{ST} = 0$ (or very small) the triplet-down state T_- is lower in energy compared to S_0 at any magnetic field $B > 0$. In consequence the singlet state will be occupied over the entire magnetic field range and we expect a line parallel to the x-axis as we see it in Fig S5e. Note that S_0 and the triplet state T_0 are energy degenerate in this case. An upper bound to this measurement can be given considering that in this sample there is a micro-magnet on top of the device (as mentioned previously in the caption of Supplementary Fig. 6). The effect of this micromagnet is to reduce the magnetic field at the center of the quantum dot by up to 0.2 T corresponding to 23 μeV . We then indicate 23 μeV as an upper bound for singlet-triplet splitting in this sample.

We have modified the caption of Supplementary Fig. 6 as following:

On top of these samples there is a micromagnet lowering the magnetic field strength at the center of the sample by up to 0.2 T corresponding to 23 μeV which is taken as a lower bound for measurable E_{ST} .

Caption of Supplementary Table 1:

Among all devices measured, in one case (data point $E_{ST} = 0 \mu\text{eV}$) we did not observe in magnetospectroscopy the signature kink associated with valley splitting. This indicates a very small valley splitting, below the lower bound of about 23 μeV set by our experimental measurement conditions

3.3 The importance of the magneto-spectroscopy data for the claim of the paper (relation to theory in Fig. 3e) and items 1 and 2 above requires that the authors should provide error bars for all measured valley splittings.

Our reply:

We thank Reviewer 3 for raising this point. We took this suggestion very seriously and we updated our fitting routine of the $(1,1) \rightarrow (2,0)$ transition. Originally, we fitted a first order polynomial to the T_- state and a zeroth order polynomial for the S_0 -state and took the crossing point of the two lines as the valley splitting. We have now implemented a more rigorous single fitting routine, which is described in Supplementary Section 1b, following the added description in Supplementary Fig. 4. We subsequently added error bars of 2σ to all E_{ST} measurements in the main text, and consequently updated the last panel of Fig 1. This was a major revision that confirms the numbers reported earlier and adds much scientific rigor and completeness to the analysis.

3.4 The surface plots in Fig. 2b and 2c span a range of 30 nm by 30 nm and APT tiles are 3 nm by 3 nm. This means the surface plot contain 10 by 10 data points. However, it seems to have 20 by 20 data points. Please resolve this inconsistency.

Our reply:

Yes, the tiles are 3x3 nm but they are only moved by 1 nm in between creating data points, so they are partially overlapping. The cubes shown in 2b and c are 20x20 nm, that is why they have 20 by 20 data points. The main reason to show these relatively small cubes in the figure is to assure that the Voronoi cells are still visible in the figure. See changes to the APT section described in point 2.3.

3.5 Reduction of the tight binding simulations to 1D is understandable as 3D is computational expensive. The use of 1D is justified by Fig. S14, however, the statistics is weak. Especially, I am not sure if this justification hold for $\rho_w > 0$ (Fig. 3g). Please comment on this issue.

Our reply:

We thank Reviewer 3 for bringing this up. We have the following changes to address this concern. First, we have expanded the analysis of the NEMO-3D simulations that were previously included in the Supplemental Information. Calculating E_v for the same heterostructures using both NEMO and the 1D two-band model, we find that the two models produce highly correlated results, which are modeled by the scaling relationship $E_v^{\text{NEMO}} = kE_v^{\text{TB}}$ for k slightly less than 1. Clearly, the 1D model is capturing most of the physics involved in valley splitting variation. These data have been added to the Supplemental Information in Section 3b.

To address concerns about using the 1D model for $\rho_w > 0$, we did some additional NEMO simulations using quantum wells with 0%, 5%, and 10% Ge in the quantum well. The resulting valley splittings increase, on average, with increasing Ge, corroborating the claims made in the paper. Additionally, we simulate each of these heterostructures using the 1D model as well, and we show again that the 1D and 3D models are highly correlated, and the relationship can be modeled as $E_v^{\text{NEMO}} = kE_v^{\text{TB}}$ for k slightly less than 1. So, even with nonzero Ge content, the 1D model captures much of the physics of the valley splitting in these devices. These new simulations can also be found in the Supplemental Information in Section 3b.

3.6 How sensitive are the calculated valley splitting on the choice of the assumed electric field in z-direction?

Our reply:

Increasing the electric field will increase the valley splitting on average, as more of the dot wavefunction penetrates high-Ge layers, thereby increasing the alloy disorder. We did some simulations of quantum wells A and B using vertical fields varying from 0 to 0.014 V/nm, showing an average valley splitting variation by a factor of 2 over the whole range of fields. Even at zero vertical field, though, there is a sizeable variation in the valley splitting due to alloy disorder. These simulations have been added to the Supplemental Information in Section 3f.

3.7 It is pointed out in the text that some APT samples have atomic step features, but these seem to be neglected by theory. Can they be neglected or is the theory merely incomplete in the sense that correlated interface disorder must be included in the future?

Our reply:

In this work, we focus on alloy disorder as the principal source of valley splitting fluctuations. Because most of the APT samples do not show steps, we focus on heterostructures without steps in this work.

However, atomic steps are an important source of variation in Si/SiGe heterostructures, and we have been studying their effect in combination with alloy disorder. We have found that, for devices with interfaces 4τ wider than about 0.4 nm, alloy disorder becomes the dominant source of E_v fluctuations, and steps play a much less important role. Because the heterostructures here have interface widths $4\tau \approx 0.8$ nm, the valley splitting fluctuations should be primarily driven by alloy disorder, not steps. In Fig. R1 below, we compare valley splitting distributions, with and without a step through the center of the dot location, for various interface widths. For interfaces wider than about 0.5 nm, the distributions become nearly identical. We are preparing a separate publication with the results of detailed studies of the interplay between interface steps and alloy disorder.

3.8 Can the presented theory explain the (monotonic) variation of the valley splitting observed in Ref 14 and arXiv:2103.14702 (latter should be cited)? There the quantum dots are moved across a small distance (10 nm). The observations were so far reasoned by atomistic steps.

Our reply:

Because the valley splitting depends on the local concentrations of Ge at a particular location in the heterostructure, we do expect the valley splitting to fluctuate as the dot is moved across the device. Fig. R2

Figure R1: Valley splitting distributions for heterostructures with (orange) and without (blue) a single ML step through the center of the dot location. Each bar shows the mean and 25-75 percentile range for 1000 valley splitting simulations using the 2D 2-band tight-binding model. X-marks indicate the mean of 10 NEMO-3D simulations. We assume isotropic dots with orbital splitting $\hbar\omega = 2$ meV and a vertical field $E = 0.005$ V/nm. The quantum wells are 80 ML wide and have interfaces parameterized by sigmoid functions. This figure demonstrates that alloy disorder dominates in determining valley splittings when the interface width 4τ exceeds ~ 0.4 nm

below shows the valley splitting fluctuations as a dot is moved across a heterostructure, which clearly shows large fluctuations as the dot is moved. So, large variability in E_v as dots sample new regions of a heterostructure can be explained by this model. Moreover, Fig. R1 suggests that a crossover between step and alloy-dominated behavior occurs at a very narrow interface width, which is not known in the experiments of Dodson et al. Therefore neither steps nor alloy disorder may be ruled out in that case. We focus more on this analysis in a future publication. Also, we have added a citation to Dodson et al., Phys. Rev. Lett. 128, 146802 in the main text.

3.9 The term control (abstract line 18) is exaggerated. The authors improved the interface quality and suggest a new approach to increase the valley splitting. However, experimental proof is not shown for the latter and therefore control stretches it too much.

Our reply:

We thank the reviewer for this comment and agree to correct for this. See answer to point 2.4 above

Figure R2: The valley splitting distribution as a function of dot position, across a 100 by 100 nm region of a heterostructure. To generate this plot, we first generate a full 180 by 180 nm heterostructure atom-by-atom. Then, for each pixel, we compute the weighted average Ge concentration at each layer for a dot centered on that pixel. We assume the dot is in the ground state of an isotropic harmonic oscillator potential with orbital splitting $\hbar\omega = 2$ meV. Then, we feed the resulting Ge concentrations at each layer into the 1D 2-band tight-binding model. We use quantum wells 80 ML wide, sigmoid interfaces with width $4\tau = 10$ ML, and a vertical field $E_z = 0.005$ V/nm. This figure demonstrates that the alloy fluctuations mechanism yields substantial changes of valley splitting when a dot is moved laterally.

References

- [1] F Borjans et al. “Single-spin relaxation in a synthetic spin-orbit field”. In: *Physical Review Applied* 11.4 (2019), p. 044063.
- [2] M. G. Borselli et al. “Measurement of valley splitting in high-symmetry Si/SiGe quantum dots”. In: *Applied Physics Letters* 98.12 (2011), p. 123118. ISSN: 00036951.
- [3] Edward H. Chen et al. “Detuning Axis Pulsed Spectroscopy of Valley-Orbital States in Si / Si - Ge Quantum Dots”. en. In: *Physical Review Applied* 15.4 (Apr. 2021), p. 044033. ISSN: 2331-7019. DOI: 10.1103/PhysRevApplied.15.044033. URL: <https://link.aps.org/doi/10.1103/PhysRevApplied.15.044033> (visited on 02/04/2022).
- [4] Ondrej Dyck et al. “Accurate quantification of Si/SiGe interface profiles via atom probe tomography”. In: *Advanced Materials Interfaces* 4.21 (2017), p. 1700622.
- [5] H Ekmel Ercan, SN Coppersmith, and Mark Friesen. “Strong electron-electron interactions in Si/SiGe quantum dots”. In: *Physical Review B* 104.23 (2021), p. 235302.
- [6] Rifat Ferdous et al. “Valley dependent anisotropic spin splitting in silicon quantum dots”. In: *npj Quantum Information* 4.1 (2018), p. 26.
- [7] Arne Hollmann et al. “Large, Tunable Valley Splitting and Single-Spin Relaxation Mechanisms in a $\text{Si}/\text{Si}_x\text{Ge}_{1-x}$ Quantum Dot”. In: *Physical Review Applied* 13.3 (Mar. 2020), p. 034068. DOI: 10.1103/PhysRevApplied.13.034068. URL: <https://link.aps.org/doi/10.1103/PhysRevApplied.13.034068> (visited on 02/04/2022).

- [8] Arne Hollmann et al. “Large, Tunable Valley Splitting and Single-Spin Relaxation Mechanisms in a Si/SiGe 1- x Quantum Dot”. In: *Physical Review Applied* 13.3 (2020), p. 034068.
- [9] E. Kawakami et al. “Electrical control of a long-lived spin qubit in a Si/SiGe quantum dot”. en. In: *Nature Nanotechnology* 9.9 (Sept. 2014), pp. 666–670. ISSN: 1748-3395. DOI: 10.1038/nnano.2014.153. URL: <https://www.nature.com/articles/nnano.2014.153> (visited on 07/08/2022).
- [10] Ruoyu Li et al. “A crossbar network for silicon quantum dot qubits”. In: *Science Advances* 4.7 (2018), eaar3960.
- [11] Thomas McJunkin et al. “SiGe quantum wells with oscillating Ge concentrations for quantum dot qubits”. In: *Preprint at http://arxiv.org/abs/2112.09765* (2021).
- [12] X Mi, S Kohler, and Jason R Petta. “Landau-Zener interferometry of valley-orbit states in Si/SiGe double quantum dots”. In: *Physical Review B* 98.16 (2018), p. 161404.
- [13] Xiao Mi et al. “High-resolution valley spectroscopy of Si quantum dots”. In: *Physical review letters* 119.17 (2017), p. 176803.
- [14] Don Monroe et al. “Comparison of mobility-limiting mechanisms in high-mobility Si_{1-x}Ge_x heterostructures”. In: *Journal of Vacuum Science & Technology B: Microelectronics and Nanometer Structures Processing, Measurement, and Phenomena* 11.4 (July 1993), pp. 1731–1737. ISSN: 1071-1023. DOI: 10.1116/1.586471. URL: <https://avs.scitation.org/doi/abs/10.1116/1.586471> (visited on 02/08/2022).
- [15] S. J. Pennycook. “Structure determination through Z-contrast microscopy”. en. In: *Advances in Imaging and Electron Physics*. Ed. by Peter W. Hawkes et al. Vol. 123. Microscopy, Spectroscopy, Holography and Crystallography with Electrons. Elsevier, Jan. 2002, pp. 173–206. DOI: 10.1016/S1076-5670(02)80063-5. URL: <https://www.sciencedirect.com/science/article/pii/S1076567002800635> (visited on 07/07/2022).
- [16] Stephan G. J. Philips et al. “Universal control of a six-qubit quantum processor in silicon”. In: arXiv:2202.09252 (Feb. 2022). arXiv:2202.09252 [cond-mat, physics:quant-ph] type: article. URL: <http://arxiv.org/abs/2202.09252> (visited on 07/07/2022).
- [17] Maximilian Russ, Csaba G Péterfalvi, and Guido Burkard. “Theory of valley-resolved spectroscopy of a Si triple quantum dot coupled to a microwave resonator”. In: *Journal of Physics: Condensed Matter* 32.16 (Apr. 2020), p. 165301. ISSN: 0953-8984, 1361-648X. DOI: 10.1088/1361-648X/ab613f. URL: <https://iopscience.iop.org/article/10.1088/1361-648X/ab613f> (visited on 07/08/2022).
- [18] P. Scarlino et al. “Dressed photon-orbital states in a quantum dot: Intervalley spin resonance”. In: *Phys. Rev. B* 95 (16 Apr. 2017), p. 165429. DOI: 10.1103/PhysRevB.95.165429. URL: <https://link.aps.org/doi/10.1103/PhysRevB.95.165429>.
- [19] Zhan Shi et al. “Tunable singlet-triplet splitting in a few-electron Si/SiGe quantum dot”. In: *Applied Physics Letters* 99.23 (2011), p. 233108.
- [20] M. L. V. Tagliaferri et al. “Impact of valley phase and splitting on readout of silicon spin qubits”. en. In: *Physical Review B* 97.24 (June 2018), p. 245412. ISSN: 2469-9950, 2469-9969. DOI: 10.1103/PhysRevB.97.245412. URL: <https://link.aps.org/doi/10.1103/PhysRevB.97.245412> (visited on 07/07/2022).
- [21] L. M. K. Vandersypen et al. “Interfacing spin qubits in quantum dots and donorshot, dense, and coherent”. In: *npj Quantum Information* 3.1 (2017), p. 34. DOI: 10.1038/s41534-017-0038-y.
- [22] Georges Voronoi. “Nouvelles applications des paramètres continus à la théorie des formes quadratiques. Deuxième mémoire. Recherches sur les paralléloèdres primitifs.” In: *Journal für die reine und angewandte Mathematik (Crelles Journal)* 1908.134 (1908), pp. 198–287. DOI: doi:10.1515/crll.1908.134.198. URL: <https://doi.org/10.1515/crll.1908.134.198>.

- [23] Georges Voronoi. “Nouvelles applications des paramètres continus à la théorie des formes quadratiques. Premier mémoire. Sur quelques propriétés des formes quadratiques positives parfaites.” In: *Journal für die reine und angewandte Mathematik (Crelles Journal)* 1908.133 (1908), pp. 97–102. DOI: doi:10.1515/crll.1908.133.97. URL: <https://doi.org/10.1515/crll.1908.133.97>.
- [24] T. F. Watson et al. “A programmable two-qubit quantum processor in silicon”. In: *Nature* 555.7698 (2018), pp. 633–637. ISSN: 14764687. DOI: 10.1038/nature25766. URL: <http://dx.doi.org/10.1038/nature25766>.
- [25] Xiao Xue et al. “Quantum logic with spin qubits crossing the surface code threshold”. In: *Nature* 601.7893 (2022), pp. 343–347.
- [26] DM Zajac et al. “A reconfigurable gate architecture for Si/SiGe quantum dots”. In: *Applied Physics Letters* 106.22 (2015), p. 223507.
- [27] Floris A. Zwanenburg et al. “Silicon quantum electronics”. In: *Reviews of Modern Physics* 85.3 (July 2013), pp. 961–1019. DOI: 10.1103/RevModPhys.85.961. URL: <https://link.aps.org/doi/10.1103/RevModPhys.85.961> (visited on 05/27/2020).

REVIEWER COMMENTS

Reviewer #1 (Remarks to the Author):

For what concerns my criticisms explained in the previous report, the new version of the Manuscript has not been updated with the exception of weakening the claim in the abstract (1) and in the introduction (2). About (1), from the original claim of "elucidate, predict and control" the Authors admitted that they do not "control" the valley splitting (the word "control" is now removed). To me, in order to correspond to their results, the claim should be further reduced to "elucidate and *statistically* predict" so to prevent the confusion in the reader that the valley splitting of an individual quantum dot is not predicted, but only in average, as correctly stated elsewhere in the abstract.

For what concerns (2), as they consider the consequences of their study to qubit yield not central in this manuscript, they declare that the recipe of increasing the valley splitting belongs to the concluding remarks ("we propose" -> "we conclude by proposing").

The Authors honestly declare that they are concentrated not on proposing a strategy to solve the problem of the qubit yield, but to characterise the effect. Unfortunately, this appears not sufficient in 2022 for publication on Nature Communication because this was sufficient in 2013, as instead suggested from their response:

"We note that on the topic of valley splitting in Si/SiGe, other

structures have been proposed but not put in practice within the same publication in Nature Communications (for example Zhang et al, Genetic design of enhanced valley splitting towards a spin qubit in silicon, Nature Communication 4,2396 (2013))."

as the field amazingly evolved in the last 10 years and, as I have already mentioned, there is literature appearing which addresses the issue:

arxiv: 2112.09765 McJunkin et al. "SiGe quantum wells with oscillating Ge concentrations for quantum dot qubits"

where as I have written "there is 1) an engineered control by a modulated concentration of the Ge 2) an experimental verification of the simulations to show that the postulated method actually works".

For the sake of the responsibility I feel by taking this decision, before submitting this second Report I have carefully considered also the Reports of the Reviewer 2 and of the Reviewer 3. I find that the criticisms 2.4 of Reviewer 2 is very strong, but, instead of presenting further evidence to address her/his comment, the Authors admit in the abstract that their results does not "explain", but just "can explain" (they add "can" to their sentence) the effect they observe:

"We find that the concentration fluctuations of Si and Ge atoms within the 3D landscape of Si/SiGe interfaces *can* explain the observed large spread of valley splitting from measurements on many quantum dot devices."

and I find that this significantly weakens the manuscript.

To conclude, unfortunately my judgement about this Manuscript to appear on Nature Communication is not changed after the amendments, i.e. I do not recommend publication. Said this, I would like to remark again that the amount of work is huge and I appreciate that the new version provides further interesting information, so I encourage the Authors to submit to a different journal so to see this work published.

Enrico Prati

Reviewer #2 (Remarks to the Author):

The revised manuscript and reviewer comments by Paquelet Wuetz, Loser, Koelling, et al do a decent job of addressing concerns raised by the reviewers and I appreciate their completeness.

Indeed, understanding the origins of valley splitting in Si/SiGe quantum dot devices is critical to the future development of the technology. This manuscript advances the field in an important way. I especially appreciated the detailed explanation of the APT data and how it goes beyond what can be extracted from HAADF-STEM.

After reviewing the responses and revised manuscript, I think it is moving closer to publication but there are outstanding issues that I recommend should be address prior to publication. These are:

1. I remain unconvinced by the argumentation that a large orbital energy (now with measurements) is sufficient for claiming that the two-electron singlet-triplet splitting is limited by the valley splitting. It is my understanding that the size of the orbital energy matters less than the asymmetry of the confining potential. This phenomena is detailed in Melnikov PRB 73, 085320 (2006), Hanson Rev Mod Phys 79, 1217 (2007), and Jones PR Applied 12, 014026 (2019). I would be convinced by data either directly measuring the valley splitting from a single electron experiment (like the pulsed gate spectroscopy in the supplement) or a measurement of both orbital energies so it is clear what the orbital asymmetry is.

a) Also of note, the authors measure an orbital energy of 4 meV in the pulsed gate experiment yet the valley splitting simulations are performed with an orbital energy of 2 meV. I would expect the simulated valley splitting distributions to change as a consequence, so if they want to compare simulated distributions to the "valley splitting" measurements I would expect they should use the orbital energy obtained from the measurement (4 meV) instead of using 2 meV.

2. Regarding the reduction in mobility due to including a uniform Ge concentration within the quantum well, the authors argue that theoretical calculations (reference 47) indicate it will not be a limiting factor. However, the first measurements of devices with Ge in the quantum well appear to have a significant mobility impact (arxiv: 2112.09765). This should at least be mentioned as it directly contradicts the statement.

3. I didn't catch this the first time, but I think some of the device concept designs in Figure 3g are likely unrealizable. In the first reading, I hadn't understood that the Ge concentration in the barriers was also increasing because the Ge offset is fixed by 25%. Is that reasonable? Don't you run into strain issues with the substrate if the barrier contains too much Ge? Also, it is unclear how much this assumption matters but it seems too much like a toy model.

4. I understand that in Figure 3d that the radial distance represents the valley splitting, but what physical consequence does the polar angle have? Is it a problem for different quantum dots to have what looks like a purely random angle even if the likelihood of a large radial distance is sufficient?

5. In the discussion section the authors state that "For no Ge in the quantum well, as consistent with our experiments, we observe significant increases in the valley splitting with increasing width." How is that consistent with the last sentence of the measurement section "We argue that quantum wells A and B have similar $E_v \pm 2\sigma$ because the electronic ground state is confined against the top interface, which is very similar in the two quantum wells."? It seems like the data says that the bottom interface width doesn't matter, so the observations do not corroborate "significant increases in the valley splitting with increasing width". It seems like the only thing the valley splitting data says are 1) the bottom interface doesn't matter and 2) the measured distributions are consistent with

valley splitting resulting from concentrations fluctuations. I would advise removing the phrase "as consistent with our experiments" as the experiments do not support that claim.

6. Minor issue: The experimental data point in figure 3g is magenta but the text says it is a green marker.

I think we are converging and I appreciate the authors improvements to the manuscript but I think there remains issues preventing publication as is.

Reviewer #3 (Remarks to the Author):

The manuscript „Atomic fluctuations lifting the energy degeneracy in Si/SiGe quantum dots“ by B. P. Wuetz, M. P. Losert and S. Koelling et al. is a significant contribution to THE main problem (the valley splitting variations) of THE most advances spin qubit system (electron spins in Si/SiGe). It is an impressive combination of material science, complex electrical measurements of local valley splitting and simulation of valley splitting. It does not lack statistics. It gives an outlook on how to increase valley splitting, which in my opinion is a strength of the manuscript. The new version of the manuscript expresses all statements with sufficient care to differentiate results and speculative outlook. This paper will trigger discussions and more work on valley splitting in the three fields (1) material characterisation, (2) electrical quantum dot measurements and (3) theory models in the future and will be cited accordingly. I leave to the judgement of the editors whether the manuscript is sufficient for the high standard of Nature Communications.

All my items were addressed very well and in my opinion, the manuscript gained considerable quality by the review process. There is only one minor item, which I recommend to change:

1. The authors improved the fitting of the valley splittings and added error bars. They also explained that the valley splitting value data point measured as zero has to be taken with care due to the presence of a micro-magnet on that specific sample. They also add a comment about this issue in the supplements in the caption of Table S1. (one full-stop is misplaced in there). The issue of the zero data point is significant and therefore I suggest to mention it in the main text in the caption of Fig 1 panel h, where this data point appears. At the current stage of the manuscript, a superficial reader perceives a zero data-point with tiny error bar in Fig. 1h

Delft, September 26, 2022

Response to Reviewer 1

We thank the Reviewer for the constructive suggestion below, which we believe fortifies our message about statistical approach to valley splitting calculations:

To me, in order to correspond to their results, the claim should be further reduced to "elucidate and *statistically* predict" so to prevent the confusion in the reader that the valley splitting of an individual quantum dot is not predicted, but only in average, as correctly stated elsewhere in the abstract.

We have implemented this suggestion in the abstract:

*"Here, we elucidate and **statistically** predict the valley splitting by the holistic integration of 3D atomic-level properties, theory and transport."*

All other comments are editorial in nature, so we leave them to the consideration of the Editor, also in view of the second round of positive reports from the Reviewer 2 and 3.

Response to Reviewer 2

We thank the Reviewer for the new comments, and for stating that the manuscript is converging towards publication. We address the latest comments point by point below:

1. I remain unconvinced by the argumentation that a large orbital energy (now with measurements) is sufficient for claiming that the two-electron singlet-triplet splitting is limited by the valley splitting. It is my understanding that the size of the orbital energy matters less than the asymmetry of the confining potential. This phenomena is detailed in Melnikov PRB 73, 085320 (2006), Hanson Rev Mod Phys 79, 1217 (2007), and Jones PR Applied 12, 014026 (2019). I would be convinced by data either directly measuring the valley splitting from a single electron experiment (like the pulsed gate spectroscopy in the supplement) or a measurement of both orbital energies so it is clear what the orbital asymmetry is. a) Also of note, the authors measure an orbital energy of 4 meV in the pulsed gate experiment yet the valley splitting simulations are performed with an orbital energy of 2 meV. I would expect the simulated valley splitting distributions to change as a consequence, so if they want to compare simulated distributions to the "valley splitting" measurements I would expect they should use the orbital energy obtained from the measurement (4 meV) instead of using 2 meV.

Our reply: The referee is asking whether the possible asymmetry in the quantum dot potential is affecting our interpretation. In our pulsed-gate experiment shown in the Supplement, the next excited state is about 4 meV above the ground state. This is either the lowest orbital excited state (i.e. the lowest of the two excited states if the quantum dot was to be elliptical) or the valley splitting itself. A valley splitting of 4 meV would be surprisingly large, so we interpret this excitation as the lowest orbital state. Since the lowest orbital splitting is greater than the valley splitting, the measured singlet-triplet splitting cannot be larger than the value of the valley splitting. As suggested by the Referee, we have redone the calculations of the valley splitting distribution using the 4.18 meV orbital splitting and amended accordingly the text, methods, and the relevant panels in Fig. 3 and in the Supplemental Figures.

We implemented the following change to capture the argument above:

*Due to small size, our dots are strongly confined with the **lowest** orbital energy much larger than E_{ST} (Supplementary Fig. 3), similar to other Si/SiGe quantum dots[1–3].*

2. Regarding the reduction in mobility due to including a uniform Ge concentration within the quantum well, the authors argue that theoretical calculations (reference 47) indicate it will not be a limiting factor. However, the first measurements of devices with Ge in the quantum well appear to have a significant mobility impact (arxiv: 2112.09765). This should at least be mentioned as it directly contradicts the statement.

3. I didn't catch this the first time, but I think some of the device concept designs in Figure 3g are likely unrealizable. In the first reading, I hadn't understood that the Ge concentration in the barriers was also increasing because the Ge offset is fixed by 25%. Is that reasonable? Don't you run into strain issues with the substrate if the barrier contains too much Ge? Also, it is unclear how much this assumption matters but it seems too much like a toy model.

Our reply: We thank Reviewer 2 for raising these two points, which we address together.

Regarding point 3, the structures are definitively feasible from a growth and strain management perspective. SiGe virtual substrates up to even 80% Ge concentration have been grown by our group and many other internationally. Furthermore, the lattice mismatch between quantum well and virtual substrate wouldn't be an issue, exactly because we would be increasing the Ge concentration in the barriers and the quantum well by the same offset. In Fig. 3g the calculations were shown initially up to 20% Ge to highlight the trend of increasing average valley splitting and spread.

Regarding point 2, indeed, a mobility decrease in arxiv2112.09765 was reported for a Si quantum well with an oscillating Ge concentration of 5% on average. As we pointed out in our manuscript, a concentration of 5% should be sufficient to achieve (on average) a substantial increase in valley splitting. In light of these considerations and to make Fig. 3g more useful in guiding future experiments, we have restricted the x -axis range to 10% Ge and added new simulations for Ge concentrations of 2.5% and 5% (also in the Supplementary). Furthermore, we implemented the following changes toward the end of the outlook section main text:

By adding Ge to the Si quantum well in small concentrations we expect to achieve on average valley splitting in excess of 100 μ eV. Early calculations from scattering theories[4] suggest that the added scattering from random alloy disorder will not be the limiting factor for mobility in current $^{28}\text{Si}/\text{SiGe}$ heterostructures. However, an approximate twofold reduction in electron mobility was recently reported when an oscillating Ge concentration of about 5% on average is incorporated in the Si quantum well[5]. We speculate that fine tuning of the Ge concentration in the quantum well will be required for enhancing the average valley splitting whilst not compromising the low-disorder potential environment, which is important for scaling to large qubit systems.

4. I understand that in Figure 3d that the radial distance represents the valley splitting, but what physical consequence does the polar angle have? Is it a problem for different quantum dots to have what looks like a purely random angle even if the likelihood of a large radial distance is sufficient?

Our reply: The valley matrix element of a single device, from Eq. (1), is the sum of a set of complex numbers, the value of which depends on both the amplitude and phase of each number in the sum. Fig. 3(d) shows the value of the sum for many realisations of a given concentration profile, and so it is natural to show the set of complex numbers that are obtained. As the referee states correctly, and as indicated by the label in the figure, the magnitude of valley splitting of a given sample is just two times the magnitude of the valley matrix element, and the polar angle doesn't have any physical consequence.

5. In the discussion section the authors state that "For no Ge in the quantum well, as consistent with our experiments, we observe significant increases in the valley splitting with increasing

width." How is that consistent with the last sentence of the measurement section "We argue that quantum wells A and B have similar $E_v \pm 2\sigma$ because the electronic ground state is confined against the top interface, which is very similar in the two quantum wells."? It seems like the data says that the bottom interface width doesn't matter, so the observations do not corroborate "significant increases in the valley splitting with increasing width". It seems like the only thing the valley splitting data says are 1) the bottom interface doesn't matter and 2) the measured distributions are consistent with valley splitting resulting from concentrations fluctuations. I would advise removing the phrase "as consistent with our experiments" as the experiments do not support that claim.

Our reply: We are very thankful for this comment. Our sentence "For no Ge in the quantum well..." was obviously poorly written and led to a clear misunderstanding. We simply wanted to say that our experiments are, indeed, performed without intentional Ge in the quantum wells. With the word "observation" we were actually referring to the results of the calculations. Indeed, for the quantum well thickness considered in our experiments, the valley splitting measurements show that the only interface that matters is the top one. For simplicity (and completeness), the model in Fig. 3g varies both interfaces width at the same time.

We reworded the sentence to fix this misunderstanding and prevent confusion arising:

For no intentional Ge in the quantum well, as consistent with the heterostructure growth profile of our measured quantum dots, the calculations show a significant increase in the valley splitting with increasing width of the interfaces."

6. Minor issue: The experimental data point in figure 3g is magenta but the text says it is a green marker.

We thank the referee for pointing this out and changed the text from: "green marker" to: "magenta marker".

Response to Reviewer 3

We thank the Reviewer for the final suggestion before publication that we address below.

...The issue of the zero data point is significant and therefore I suggest to mention it in the main text in the caption of Fig 1 panel h, where this data point appears. At the current stage of the manuscript, a superficial reader perceives a zero data-point with tiny error bar in Fig. 1h

Our reply: Yes, we agree and have added the following sentence in the caption of Fig 1h:

"For quantum well B, the data point $E_V = 0 \mu\text{eV}$ indicates that the kink in magnetospectroscopy associated with valley splitting was not observed and, consequently, that the valley splitting is below the lower bound of about $23 \mu\text{eV}$ set by our experimental measurement conditions (see Supplementary Fig. 6 and Supplementary Table 1)."

References

- ¹D. Zajac, T. Hazard, X. Mi, K. Wang, and J. R. Petta, "A reconfigurable gate architecture for Si/SiGe quantum dots", Applied Physics Letters **106**, 223507 (2015).
- ²X. Mi, C. G. Péterfalvi, G. Burkard, and J. R. Petta, "High-resolution valley spectroscopy of Si quantum dots", Physical Review Letters **119**, 176803 (2017).

- ³A. Hollmann, T. Struck, V. Langrock, A. Schmidbauer, F. Schauer, T. Leonhardt, K. Sawano, H. Riemann, N. V. Abrosimov, D. Bougeard, and L. R. Schreiber, “Large, Tunable Valley Splitting and Single-Spin Relaxation Mechanisms in a Si/Si_{1-x}Ge_x Quantum Dot”, *Physical Review Applied* **13**, 034068 (2020).
- ⁴D. Monroe, Y. H. Xie, E. A. Fitzgerald, P. J. Silverman, and G. P. Watson, “Comparison of mobility limiting mechanisms in high mobility Si_{1-x}Ge_x heterostructures”, *Journal of Vacuum Science & Technology B: Microelectronics and Nanometer Structures Processing, Measurement, and Phenomena* **11**, 1731–1737 (1993).
- ⁵T. McJunkin, B. Harpt, Y. Feng, M. Losert, R. Rahman, J. P. Dodson, M. A. Wolfe, D. E. Savage, M. G. Lagally, S. N. Coppersmith, M. Friesen, R. Joynt, and M. A. Eriksson, “SiGe quantum wells with oscillating Ge concentrations for quantum dot qubits”, Preprint at <http://arxiv.org/abs/2112.09765> (2021).

REVIEWERS' COMMENTS

Reviewer #2 (Remarks to the Author):

I find the manuscript by Paquelet Wuetz et al to be sufficiently improved so as to support publication. I believe they have adequately addressed all of the outstanding issues and the work should proceed to publication. I also believe this work to be of high value to the Si quantum dot spin qubit community and look forward to its widespread dissemination.